# *Drosophila* USP22/nonstop polarizes the actin cytoskeleton during collective border cell migration

Hammed Badmos[1,2] , Neville Cobbe[1] , Amy Campbell[1] , Richard Jackson[3], and Daimark Bennett[1,2]

Polarization of the actin cytoskeleton is vital for the collective migration of cells in vivo. During invasive border cell migration in *Drosophila*, actin polarization is directly controlled by the Hippo signaling complex, which resides at contacts between border cells in the cluster. Here, we identify, in a genetic screen for deubiquitinating enzymes involved in border cell migration, an essential role for nonstop/USP22 in the expression of Hippo pathway components *expanded* and *merlin*. Loss of *nonstop* function consequently leads to a redistribution of F-actin and the polarity determinant Crumbs, loss of polarized actin protrusions, and tumbling of the border cell cluster. Nonstop is a component of the Spt-Ada-Gcn5-acetyltransferase (SAGA) transcriptional coactivator complex, but SAGA's histone acetyltransferase module, which does not bind to *expanded* or *merlin*, is dispensable for migration. Taken together, our results uncover novel roles for SAGA-independent nonstop/USP22 in collective cell migration, which may help guide studies in other systems where USP22 is necessary for cell motility and invasion.

## Introduction

Tightly regulated cell migration is vital for normal development, and aberrant migration is involved in a number of human diseases, including cancer metastasis, inflammatory diseases, and various birth abnormalities (Schumacher, 2019; Stuelten et al., 2018). In many instances, cells move by the process of collective migration in vivo, whereby migratory cells remain connected by cell–cell junctions, and show group polarization and coordinated cytoskeletal dynamics (Haeger et al., 2015; Mishra et al., 2019; Norden and Lecaudey, 2019). This mode of migration is exemplified by the movement of border cells (BCs) in *Drosophila melanogaster* (Video 1). In this process, a cluster of five to eight cells are recruited from the follicular epithelium in the ovary by a pair of nonmotile polar cells. Both cell types migrate as a cluster from the anterior basal lamina of the egg chamber, invading the underlying germ line, to the anterior border of the oocyte where they are involved in patterning before egg fertilization (Montell et al., 2012).

Studies of this process over the past 20 yr have identified key features of the genetic program required for BC migration that control the specification of the migratory cluster (Bai et al., 2000; Montell et al., 1992; Silver and Montell, 2001), organization of cluster polarity and detachment from the epithelium (Abdelilah-Seyfried et al., 2003; McDonald et al., 2008; Pinheiro and Montell, 2004), timing of migration (Godt and Tepass, 2009; Jang et al., 2009), adhesion of the cluster (Cai et al., 2014;

Niewiadomska et al., 1999), and guidance to the oocyte (Bianco et al., 2007; Duchek and Rørth, 2001; Duchek et al., 2001; McDonald et al., 2003). Details have also emerged regarding the dynamic organization of the actin cytoskeleton, which is an essential driver of this process (Plutoni et al., 2019), with recent studies identifying an important role for the Hippo pathway in linking determinants of cell polarity with polarization of the actin cytoskeleton in migrating clusters (Lucas et al., 2013). Our understanding of the interplay between polarity determinants and the actin cytoskeleton, however, remain incomplete, as does knowledge of the regulatory networks responsible for first establishing this polarity.

Ubiquitination of proteins by ubiquitin E3 ligases and removal by deubiquitinating enzymes (DUBs) regulate a raft of intracellular functions from protein stability and enzyme activity to receptor internalization and protein–protein interactions (Clague et al., 2013; Swatek and Komander, 2016). There is a growing body of evidence that ubiquitination plays a role in regulating the motility of single cells in culture (Cai et al., 2018), but little is known about its contribution to collective migration in vivo. Here, we report our identification of nonstop (*not*) from a screen of DUBs involved in BC migration. *not* encodes the *Drosophila* USP22 orthologue (Martin et al., 1995) and is best known as the enzymatic component of the histone H2B DUB module of the Spt-Ada-Gcn5-acetyltransferase (SAGA)

[1]Department of Biochemistry, Institute of Integrative Biology, University of Liverpool, Liverpool, UK;   [2]Department of Molecular Physiology and Cell Signalling, Institute of Systems, Molecular and Integrative Biology, University of Liverpool, Liverpool, UK;   [3]Liverpool Clinical Trials Centre, University of Liverpool, Liverpool, UK.

Correspondence to Daimark Bennett: daimark@liverpool.ac.uk.



transcriptional coactivator complex (Koutelou et al., 2010; Lee et al., 2011; Zhang et al., 2008). Histone modifications such as acetylation and ubiquitination modulate the accessibility of genomic loci to transcriptional machinery, with ubiquitination being associated with both activation and repression (Weake and Workman, 2008). Correspondingly, SAGA is associated with the enhancers, promoters, and sites of paused RNA polymerase II at genes in multiple tissues during *Drosophila* embryogenesis, and Not activity within SAGA is required for full expression of tissue-specific genes (Weake et al., 2011).

*not*/USP22 plays essential roles during embryogenesis in *Drosophila* and mammals (Li et al., 2017; Lin et al., 2012) as well as in neural development (Weake et al., 2008) and lineage specification (Kosinsky et al., 2015). In the *Drosophila* nervous system, *not* loss of function is associated with defects in the migration of a subset of glial cells to their appropriate position in the developing optic lobe and subsequent targeting of photoreceptor axons in the lamina (Martin et al., 1995; Poeck et al., 2001). The underlying mechanisms are not fully understood, but it has recently been suggested that this role may be mediated in part by a SAGA-independent role of Not in deubiquitinating and stabilizing the actin regulator Scar (Cloud et al., 2019). Here, we find that in collective BC migration, *not* functions independently of both Scar and SAGA to regulate the expression of two upstream components of the Hippo pathway, resulting in the loss of F-actin polarity, the mislocalization of polarity determinants, a change in the size and orientation of cellular protrusions, and the loss of polarized migration, identifying *not* as an important regulator of the transcriptional network underlying collective cell migration.

## Results

### *not* is required for invasive BC migration

We identified *not* in an RNAi screen for DUBs required for BC migration. Expression of transgenic inverted repeat constructs for *not* (*not^IR*) in the outer BCs severely delayed BC migration compared with controls (Fig. 1, A–E). In many cases, the BC cluster failed to detach from the follicular epithelium (e.g., Video 2). These migration defects were significantly rescued by a full-length synthetic RNAi-resistant transgene (*not^+r*, see Materials and methods), confirming the requirement for *not* (Fig. 1 E). Complementation experiments with an extra UAS transgene did not have the same effect, verifying that partial rescue was not simply a result of titrating the GAL4 and reducing the expression of double-strand RNA (dsRNA; Fig. 1 E). Expression of *not^+r* alone in outer BCs had no effect on migration (Fig. 1, C and E). To further confirm *not*'s role in BC migration, we generated homozygous clones for an amorphic *not* allele (*not^1*). Notably, BC clusters genetically mosaic for *not^1* showed greatly retarded migration (Fig. 1, F–H), with severity being dependent on the proportion of mutant cells in the cluster (Fig. 1 H). These defects were fully rescued by transgenic expression of *not^+r* (Fig. 1 H). Clusters containing only *not^1* outer BCs showed strong migration defects (Fig. 1, G and H), and this was not significantly enhanced by the presence of *not^1* polar cells (compare >50% or >80% *not^1* outer BC with >50% and 100% *not^1* cluster, respectively; Fig. 1 H), suggesting that *not* is dispensable in polar cells for migration of

the cluster. To further confirm the requirement for *not* in outer BCs, we used a CRISPR-Cas-9 approach (Port et al., 2014); expression of single-guide RNA (sgRNA) for *not* (*not^sgRNA*) along with *Cas-9* (Port and Bullock, 2016) in all cells in the cluster strongly abrogated migration; expression in outer BCs from a later stage of development had a less pronounced effect, whereas expression solely in polar cells had no effect (Fig. S1 A). Unlike in controls, splitting of BC clusters was observed in 28% of stage 9 or 10 *not^1* egg chambers (Fig. 1, I and J), indicative of a defect in maintaining the integrity of BC–BC contact. Splitting was observed in clusters containing WT polar cells (Figs. 1 I and 2 A), suggesting that this effect was cell autonomous to outer BCs. Interestingly, split parts of the cluster were observed to contain both WT and *not^1* cells (Fig. 1 I) or just *not^1* cells (Fig. 2 C), suggesting that mutant cells do not have an intrinsic difference in affinity to each other or WT cells. Taken together, these data identify *not* as a novel regulator of BC migration.

### *not* regulates polar cell number

At stage 8 of oogenesis, a pair of anterior polar cells secrete unpaired (Upd) ligand, which activates the Janus kinase-signal transducer and activation of transcription (JAK-STAT) signaling pathway in surrounding follicle cells, leading to the recruitment of five to eight follicle cells into a migratory cluster (Beccari et al., 2002; Silver and Montell, 2001). When we looked at upstream signaling using *upd-lacZ*, we observed that 32% of *not^1* clusters possessed more than two *upd-lacZ* positive polar cells (Fig. 2, A and B). *not^1* clusters also contained on average a 1.7-fold higher number of BCs than controls, and this was correlated with the number of *upd-lacZ*–positive polar cells (Fig. 2 B), suggesting that the presence of additional polar cells led to the recruitment of additional BCs into the cluster. However, extra polar cells occurred in clusters containing WT polar cells (Fig. 2, C and D), indicating that the effect on polar cell number may be noncell autonomous. Targeting *not* in polar or outer BCs by CRISPR-Cas9 did not significantly affect cell number (Fig. S1 B). This may be because of differences in the efficacy of inducing loss of function or because there is a requirement for *not* at an earlier point in egg chamber development that is disrupted in *not^1* clones. The presence of extra BCs might affect function. However, strong migration defects were clearly observed in *not^1* mutant clusters with normal numbers of cells (six to eight cells; Fig. 2 E); at best, there was a weak negative correlation between total number of nuclei and percent migration (R = −0.43). Cell splitting was also not limited to larger clones containing extra cells (Fig. 2 F), and there was not a strong correlation between cluster size and frequency of splitting (R = 0.01). To explore the requirement for *not* in BC signaling, we looked at the expression of *slbo*, a downstream target of Upd-JAK-STAT signaling in the migratory outer BCs. The level of *slbo-lacZ* was not significantly different between *not^1* cells and WT siblings within mosaic BC clusters (Fig. 2, G–I). *not* was also not required for the expression pattern of Eyes Absent protein (Fig. 2, J and K), which is expressed in outer BCs to repress polar cell fate in these cells (Bai and Montell, 2002). Therefore, we conclude that *not* does not affect the expression levels of genes in migratory outer BCs that specify their fate.

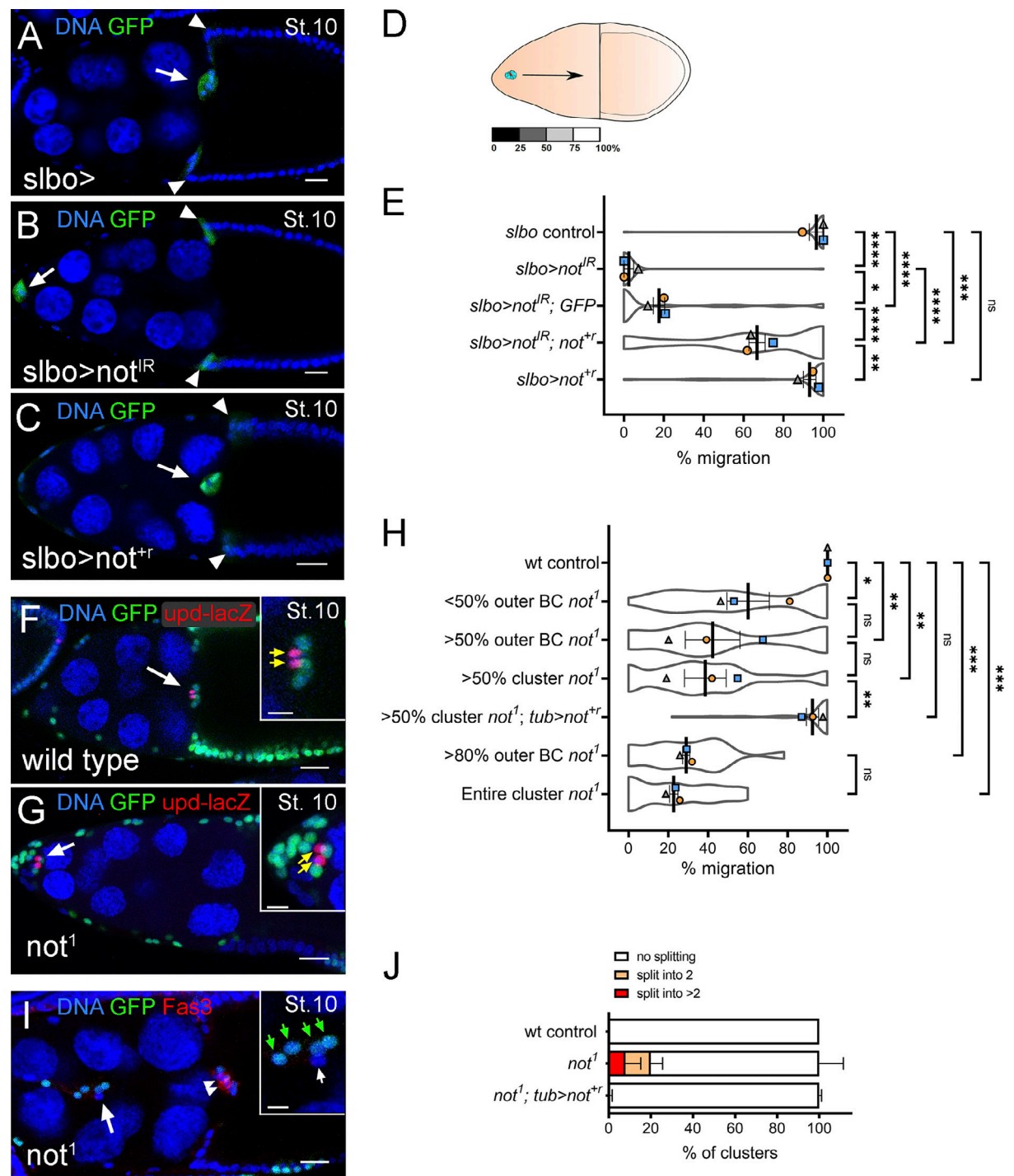

Figure 1. ***not* is required in outer BCs for migration and cluster integrity. (A–D)** Confocal micrographs of stage 10 egg chambers: outer BCs (arrows) and some centripetal follicle cells (arrowhead) marked with GFP (green) under the control of *slbo-GAL4*. **(A)** *slbo-GAL4* control (*slbo>*) showing complete migration of the BC cluster. **(B and C)** RNAi knockdown of *not* (*slbo>not^IR*) abrogated BC migration, whereas overexpression of *not^+r* (C; *slbo>not^+r*) did not. **(D)** Depiction of egg chamber showing distance measured for quantification of migration. **(E)** Plots showing mean migration ± SEM derived from means of each replicate superimposed on violin plots of migration measurements for each indicated genotype. The effect of *not^IR* can be partially rescued by overexpression of RNAi-resistant *not* (*slbo>not^+r not^IR*) but not overexpression of UAS-GFP (*slbo>not^IR*, GFP). *, P < 0.05; **, P < 0.01; ***, P < 0.001; ****, P < 0.0001 by one-way ANOVA. **(F and G)** Confocal micrographs of egg chambers with GFP-labeled clones (green); polar cells are marked with *upd-lacZ* (red; indicated with yellow arrows in inset). Compared with control BC clusters, which routinely complete migration at stage 10 (F), *not^1* clusters display defective migration (G). **(H)** Quantitation of migration defects at stage 10 reveals that clusters with >50% mutant outer BCs are more severely affected than those with <50% mutant outer BCs; migration is largely restored by *not^+r* overexpression (*tub>not^+r*; *not^1*). Migration defect in clusters with >80% mutant outer BCs was not significantly different from those in which the entire cluster was mutant. **(I)** Stage 10 egg chamber with polar cells (arrowheads) labeled with Fas3 (in red) showing splitting of *not^1* BCs. The inset shows a magnified image of GFP+ BCs (green arrows, GFP−; white arrows, BCs). **(J)** 18% of clusters displayed splitting into two groups of cells, 10% of clusters split into more than two groups of cells. Nuclei are labeled with TO-PRO-3 (blue) in all images. Replicate trials consisted of ≥20 egg chambers per genotype. Scale bars in confocal images are 25 µm (insets, 10 µm). St., stage.

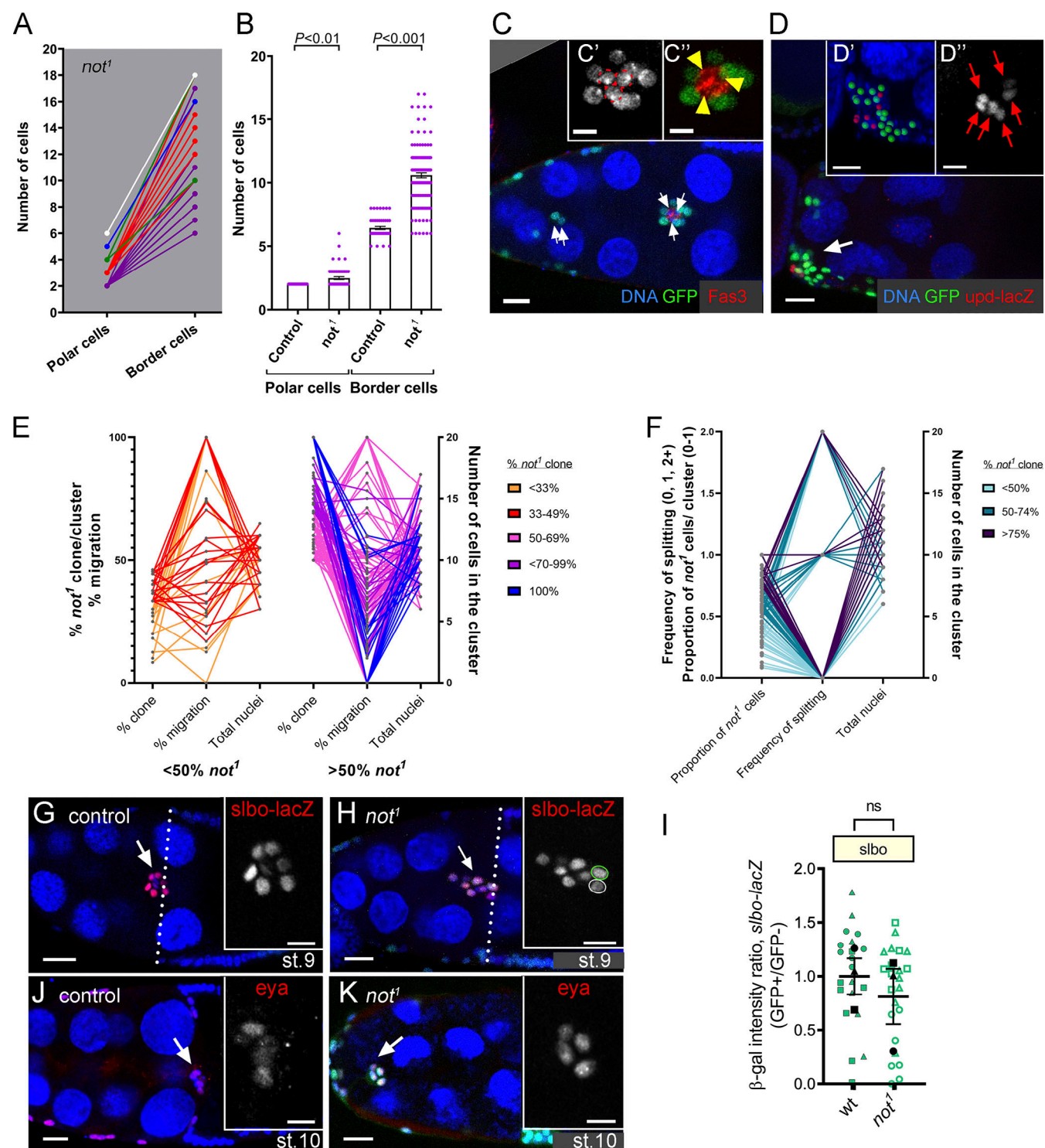

Figure 2. ***not* regulates polar cell number and cluster size. (A)** Graph of number of *upd-lacZ*[+] polar cells versus BCs in individual *not1* BC clusters colored according to polar cell number: two, purple (*n* = 36); three, red (*n* = 12); four, green (*n* = 3); five, blue (*n* = 1); six, white (*n* = 1). **(B)** Quantitation of polar and BC numbers reveals a significant increase in numbers of both *upd-lacZ*[+] polar cells and BCs in *not1* clusters compared with controls. **(C)** Stage 10 egg chamber with *not1* BCs labeled by MARCM with GFP (green), surrounding WT polar cells labeled with anti-Fasciclin 3 staining (Fas3; red). Lagging *not1* BCs can be seen (arrows). Insets show magnified image of BC with three polar cells (C', nuclei indicated with dotted red lines; C", Fas3 staining in red). **(D)** *not1* BC cluster containing six *upd-lacZ*[+]–labeled nuclei. Insets shows magnified image segmented in 3D into *upd-lacZ*[+] nuclei (red spots) and GFP *not1* cells (green; D') or *upd-lacZ* expression alone (grayscale; D"). **(E)** Graph showing relationship between percent *not1* clones/cluster, number of cells in the cluster, and percent migration for individual sets of measurements. **(F)** Graph showing relationship between proportion of *not1* clones/cluster, number of cells in the cluster, and frequency of splitting. In E and F, color coding refers to percentage of the cluster that was mutant. **(G)** Control egg chamber at mid-stage 9 showing slow BC expression with the *slbo-lacZ* reporter (red) in migrating BCs. Inset shows magnified image of *slbo-lacZ* alone (grayscale). **(H)** Stage 10 egg chamber with *not1* BCs labeled by MARCM with GFP (green). Inset shows that the normal pattern of *slbo-lacZ* is detected in GFP mutant cells (green outline) compared with WT sibling (white

outline). **(I)** Quantitation of relative *slbo-lacZ* signal intensity (GFP⁻, internal control: GFP⁺ homozygous sibling cell; see Materials and methods), showing no significant difference in *slbo* expression between WT and *not¹* cells. Plots show overall mean ± SEM derived from means of *n* > 7 egg chambers/replicate superimposed on Beeswarm plots of individual measurements for each of the indicated genotypes. **(J)** Control stage 10 egg chamber showing anti–Eyes Absent antibody staining (Eya, red). Inset shows magnified image of Eya alone (grayscale). **(K)** Stage 10 egg chamber with GFP-labeled *not¹* BCs (green). In both control and *not¹* cluster cells, Eya is restricted to outer BCs. Nuclei are labeled with TO-PRO-3 (blue) in all images. Scale bars in confocal images are 25 µm (insets, 10 µm). st., stage.

### *not* is required for normal actin polarity in migratory BCs

Following their specification, BCs undergo two phases of collective cell migration (Bianco et al., 2007). In the initial phase, leader cells exhibit long, highly polarized F-actin protrusions that are required for adhesion to and migration through the substratum (Fulga and Rørth, 2002). Later, F-actin accumulates around the cortex of the cluster as cells alternate their position in the cluster (Bianco et al., 2007). In *not¹* clones, we saw loss of initial F-actin polarity, and F-actin accumulated along BC–BC junctions (Fig. 3, A and B). There was a 2.6-fold shift in relative distribution of F-actin toward the interior BC junctions in *not¹* clusters compared with controls (Fig. 3 C). This change in distribution was rescued by *not⁺ʳ* overexpression (Fig. 3 C). We similarly observed a disruption to the distribution of F-actin when we targeted *not* by CRISPR-Cas9 (Fig. S1, C and D). By live imaging, we found that progressive migration was reduced by 80% from 0.45 µm/min in controls to 0.09 µm/min in *not¹* BC clusters (P < 0.01). This was accompanied by loss of initial F-actin polarity (Fig. 4, A and B; compare with Videos 3 and 4) and tumbling motion (Fig. 4, A–C; see Materials and methods). Further analysis revealed that while there was not a global reduction in the number of protrusions in *not¹* clusters (Fig. 4 D), there was a significant change in the distribution of the number (Fig. 4 E) and size (Fig. 4, F and G) of protrusions from a front bias in controls (54% of protrusions) to the sides (63% of protrusions) in *not¹* (Fig. 4 E; P < 0.01), consistent with a failure of these clusters to move in a polarized fashion. Since *not¹* clusters often failed to delaminate, we cannot rule out that some differences in protrusion dynamics result from a failure to detach from the epithelium, although we did observe similar protrusion behavior in a detached cluster that we imaged (Video 5).

### *not* does not function during BC migration by stabilizing Scar

Recent data suggest that Not is capable of interacting with Arp2/3 and the WAVE regulatory complexes (WRCs) in the cytoplasm to prevent polyubiquitination and subsequent proteasomal degradation of the WRC subunit Scar (Cloud et al., 2019). Scar/WAVE–Arp2/3 interactions result in nucleation of branched actin filament networks and in that way, regulate migration (Buracco et al., 2019; Krause and Gautreau, 2014). This prompted us to test whether loss of *not* function resulted in destabilization of Scar levels in BCs. Endogenous Scar staining was very faint (Fig. 5, A and B) compared with ectopically overexpressed Scar (Fig. 5 C), but there was no significant difference in Scar levels in *not¹* follicle cells and their heterozygous siblings (Fig. 5 D). There was a significant, though modest, enrichment of Scar in the junctions between outer BCs of *not¹* clusters accompanying the increased F-actin at this location (Fig. 5 E). Scar did not show this distribution when overexpressed (Fig. 5 C), and overexpression of Scar disrupted neither

F-actin polarity nor cell migration (Fig. 5, F–H). To test whether *scar* loss of function phenocopied *not¹* clusters, we generated homozygous clones for an amorphic *scar* allele, *scar^Δ37^* (Zallen et al., 2002). Migration of *scar^Δ37^* clusters was retarded (Fig. 5 G). However, we found that F-actin polarity was unaffected, with F-actin being predominantly distributed at the cortex of *scar^Δ37^* clusters (Fig. 5, F and I). Previous live imaging analysis of *scar* RNAi (*scar^IR^*) clusters revealed that *scar* loss of function resulted in a reduction in the number of cellular protrusions, with a higher proportion of protrusions at the rear of the cluster and fewer in the front and middle compared with controls (Law et al., 2013). These phenotypes are consistent with a reduction in migration but not with the *not¹* phenotypes described above (Figs. 3 and 4). Polarization of the polarity determinant Crb was also normal in *scar* mutant clones, suggesting that the architecture of the clusters was unaffected (Fig. 5, F and J). Taken together, we conclude that in BCs, *not* does not drive collective migration through Scar stabilization. Given the observations above, we also tested an alternative possibility that *scar* may contribute to F-actin accumulation between *not¹* outer BCs. We observed a weak suppression of *not¹*-induced defects in F-actin polarity and BC migration with *scar^IR^* (Fig. 5, F, G, and K). In summary, rather than being depleted in *not¹* clusters, Scar is slightly enriched in BC–BC junctions where it contributes to F-actin formation.

### *not* is required for the normal level and/or distribution of Hippo signaling components in BCs

The loss of normal actin polarity and early tumbling of the BC cluster are features of Hippo signaling loss of function (Lin et al., 2014; Lucas et al., 2013). In outer BCs, the key upstream components of the Hippo pathway (Crumbs [Crb], Kibra [Kib], Expanded [Ex], Merlin [Mer]) are found at sites of BC–BC contact (Lucas et al., 2013; Niewiadomska et al., 1999), where the pathway acts independently of the canonical downstream effector Yorkie to limit the activity, but not the recruitment, of the actin polymerization protein Enabled (Lucas et al., 2013). Therefore we tested whether *not* may be required for the normal level or distribution of Hippo signaling components in outer BCs. Using a transcriptional reporter of *ex* expression (*ex-lacZ*), we found a 1.7-fold reduction in *ex* levels in *not¹* cells compared with heterozygous sister cells in mosaic BC clusters (Fig. 6, A–C). Similarly, we saw a reduction in Merlin protein levels at BC–BC junctions in *not¹* cells (Fig. 6, D and E). To quantify this, we measured the effect of *not¹* on an endogenously expressed YFP-tagged Merlin transgene, which revealed a 1.6-fold reduction in YFP-Merlin levels (Fig. 6, F–H). The distribution of Enabled was largely unaffected in *not¹* clusters (Fig. 6, I–K). In follicle cells, *ex* and *mer* are redundantly required for normal localization of the

**Figure 3.** **_not_ is required for normal actin polarity in migratory BCs.** **(A)** Confocal micrographs of egg chambers harboring WT, _not¹_, or rescued _not¹_ GFP-labeled clones (_not¹; tub>not⁺ʳ_) labeled with phalloidin to visualize F-actin (red) and TO-PRO-3 to visualize nuclei (blue). Egg chambers are stage 10 except a WT control, which is shown at mid-migration at stage 9 (dotted line indicates expected position of the cluster at this stage of migration). BC clusters are indicated with arrows. In WT, F-actin is normally polarized, with high levels around the cortex, at BC–nurse cell junctions. In contrast, in _not¹_ clusters, F-actin predominantly accumulates at internal BC–BC junctions; this is rescued by transgenic overexpression of _not⁺ʳ_. Scale bars are 25 μm (red, green, and blue images) and 10 μm for magnified grayscale images of F-actin. **(B)** Representative line scans (indicated with yellow line in A) of the same genotypes showing signal intensities of F-actin from anterior (left) to posterior (right) and the change in F-actin profile in _not¹_ clusters. **(C)** Dot plots of area under curve for back, middle, and front (BMF) of the BC cluster derived from line scans through the cluster, with mean ± SEM derived from means of _n_ > 5 egg chambers/replicate, showing a consistent defect in F-actin polarization in _not¹_ clusters. F-actin polarization was restored by _not⁺ʳ_ overexpression (_tub>not⁺ʳ; not¹_). ****, P < 0.0001 by one-way ANOVA. st., stage.

apical transmembrane protein Crb (Aguilar-Aragon et al., 2020; Fletcher et al., 2012). When we examined the distribution of Crb, we found that rather than being distributed in the junctions between neighboring BCs (Niewiadomska et al., 1999), it was localized around the cortex of the cluster at the interface between BCs and nurse cells (Fig. 6, K–M). Neither the distribution nor the levels of Moesin (Moe), which stabilizes Crb at the apical membrane of epithelia by linking Crb to cortical actin (Médina et al., 2002), were affected in _not¹_ BCs (Fig. S2). Crb is required for polarization of other polarity determinants, including aPKC, in BCs (Wang et al., 2018). Correspondingly, the distribution of aPKC was also significantly disrupted in _not¹_ cells (Fig. 6, N–P). We did not observe a quantitative change in the distribution of the adherens junction protein Armadillo/β-catenin (Arm; Fig. 6,

P–R). Taken together, these data show that _not_ is required for expression of Hippo signaling components and correct recruitment of polarity determinants in outer BCs.

### _ex_ and _mer_ are targets of Not, but not the histone acetyltransferase (HAT) module of SAGA, which is dispensable for BC migration

A key and highly conserved role of Not/USP22 is to regulate gene expression, acting as a central component in the DUB module of the SAGA complex (Lee et al., 2011). By exploiting genome-wide chromatin immunoprecipitation sequencing (ChIP-Seq) data from a recent study of the _Drosophila_ SAGA complex (Li et al., 2017), we asked whether any of the canonical Hippo signaling components are transcriptional targets of Not. We found that

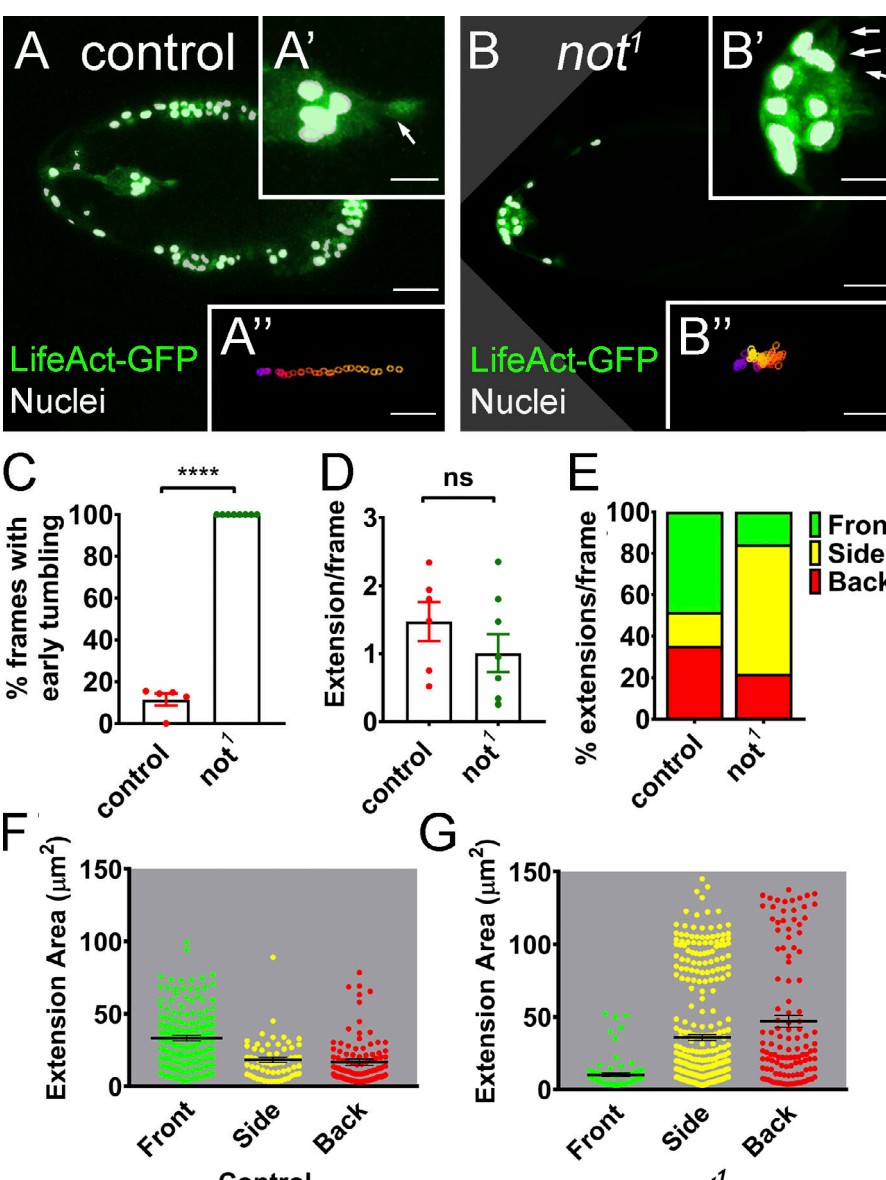

Figure 4. **Loss of *not* results in loss of polarized protrusions, retarded migration, and early tumbling. (A and B)** Still images from time-lapse imaging of LifeAct-GFP–labeled BCs near the start of BC migration with nuclear GFP-labeled MARCM clones labeled in white and LifeAct-GFP in green. **(A)** Control egg chamber with clearly visible polarized F-actin protrusion at the leading edge of the cluster (arrow in magnified image A'), leading to progressive migration from anterior to posterior (track A'', generated using a custom macro; Poukkula et al., 2011). **(B)** In contrast, *not1* clusters display multiple shorter protrusions at different positions around the cluster (arrows in B'), leading to poorly directed movement of the cluster toward the posterior pole (track B''). Scale bars in confocal images are 25 µm (10 µm for A'/B' insets and 25 µm for A''/B'' insets). **(C–G)** Quantitation of time-lapse images from WT control (n = 5) and *not1* (n = 7 clusters that failed to readily delaminate) LifeAct-GFP–labeled BC clusters, showing effects on tumbling and actin-based cellular protrusions. **(C)** Graph showing percentage of frames from the first half of migration with tumbling BCs. Individual data points together with mean ± SEM *not1* show significant increases in early tumbling. ****, P < 0.0001 by Student's *t* test. **(D)** Graph of total cellular extensions/ frame after segmentation. There is no significant difference (by Student's *t* test) between WT control and *not1*. **(E)** Graph of percentage of extensions/frame at front, back, and sides of the cluster, showing a higher proportion of extensions at the side of *not1* clusters compared with controls. **(F and G)** Measurements of the area of extensions detected at front, back, and sides of WT and *not1* clusters together with mean area ± SEM show that the size of protrusions at the front is reduced in *not1* clusters concomitantly with an increase in the size of extensions at the side and back.

there are Not binding sites in many of the Hippo pathway gene promoters, including *ex* and *mer* (Fig. 7, A and B), *hpo, kib, zyx,* and *crb* (Li et al., 2017; Table S1). Interestingly, Ada2b, a SAGA-specific HAT module subunit that anchors the HAT module to SAGA and is required for its HAT activity (Kusch et al., 2003; Lee et al., 2011; Muratoglu et al., 2003; Pankotai et al., 2005; Zsindely et al., 2009), was also reported to bind *hpo, kib, zyx,* and *crb* but not *ex* or *mer* (n = 4; Fig. 7, A and B), suggesting that *ex* and *mer* promoters are DUB-specific targets. In *ada2b* mutant clones, we saw a significant, though modest, increase in *ex-lacZ* levels (Fig. 7, C and D) but normally localized F-actin (Fig. 7, E and F); migration was significantly, though modestly, affected (mean migration 82.3 ± 3.6%; Fig. 7 G). RNAi-mediated knockdown of other HAT module components *ada3, sgf29,* or *gcn5* did not impair migration (Fig. 7 G). Knockdown of the HAT itself, encoded by *gcn5*, also had no effect on F-actin or Crb distribution (Fig. 7, F, H, and I). Taken together, these data indicate that the DUB module can regulate BC migration independently of the HAT

module. In some contexts, Yki activates the expression *ex* and *mer* and associates with chromatin-associated factors to regulate transcription (Hillmer and Link, 2019). However, consistent with previous studies, knockdown of *yki* in outer BCs did not impair migration (Fig. S3 A; Lin et al., 2014) or affect the distribution of F-actin or Crb (Fig. S3, B–D), suggesting that in BCs, Yki is not involved in recruiting Not to the promoters of *ex* and *mer*.

### Overexpression of *ex* partially rescues cell migration and polarity defects

To further explore the functional significance of reduced *ex* levels, we examined the effect of *ex* loss of function on BC polarity and migration (Fig. 8). We found that BCs mutant for an *ex* loss-of-function allele (*ex^e1*) phenocopied the effect of *not1*, albeit more weakly (Fig. 8, A–F), with a significant loss of cortical F-actin staining and a disruption of Crb distribution (Fig. 8, K and L) accompanied by abrogated migration (Fig. 8 M). Strikingly, *ex* overexpression (*ex^+*) substantially restored more

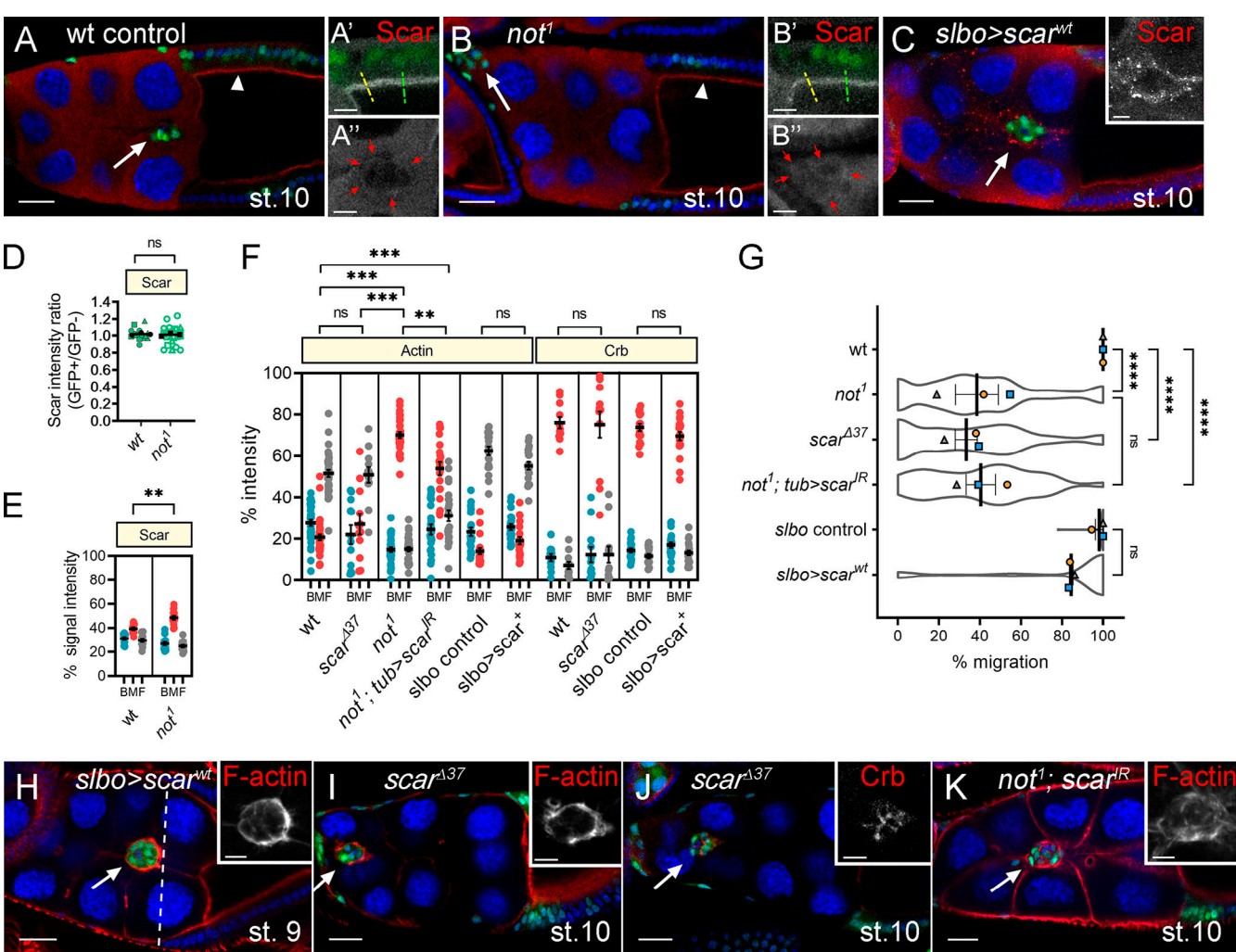

Figure 5. **Not does not act by stabilizing Scar during BC migration. (A and B)** Scar levels are not reduced in *not[1]* follicle or BCs. Confocal micrographs show Scar staining (red) in WT (A) and *not[1]* (B) GFP-labeled MARCM clones (green) at stage 10 of egg chamber development. Arrowhead, follicular epithelium; arrow, BC. Magnified image of Scar, shown in grayscale in A', predominantly localizes to apical junctions of columnar follicle cells. Magnified image in A' shows Scar staining is cytoplasmic in nurse cells but in BCs, can be detected at outer junctions of the cluster (red arrows). **(B)** Scar is similarly localized in *not[1]* clones, with no reduction in level at the apical side of follicle cells (B') but can be observed at BC–BC junctions (B''). **(C)** Overexpression of WT Scar (Scar[wt]) using *slbo-GAL4* results in a robust signal, confirming Scar staining at the outer junctions of BCs. **(D)** Quantification of Scar levels (as in A' and B'; *n* ≥ 5 egg chambers/replicate). **(E)** Quantification of Scar signal intensity at back, middle, and front (BMF) of the cluster (*n* ≥ 5 egg chambers/replicate). **(F)** Dot plots showing quantification of F-actin and Crb signal intensities across the cluster in indicated genotypes (*n* ≥ 3 cluster/replicate). **(G)** Graphs showing mean migration ± SEM derived from means of *n* > 15 egg chambers/replicate of the indicated genotypes superimposed on violin plots. **(H and I)** GFP-labeled BC clusters (in green) with overexpressed Scar[wt] or *scar* loss of function (*scar[Δ37]*), respectively, show normal F-actin polarity. **(J)** GFP-labeled *scar[Δ37]* clusters (in green) show normal Crb polarity. **(K)** Accumulation of F-actin into the junction in *not[1]* clones is partially rescued by knockdown of *scar* (*not[1] scar[IR]*). TO-PRO-3 (blue) stains all nuclei in all images. Scale bars are 25 µm (inset, 10 µm). **, P < 0.01; ***, P < 0.001; ****, P < 0.0001. st., stage.

normal Crb and F-actin distributions in *not[1]* BCs (Fig. 8, G, H, K, and L). *ex[+]* also partially suppressed the effect of *not[1]* on migration, although this was not statistically significant (mean percent migration of *ex[+] not[1]* BC clusters, 57.4 ± 3.2% compared with 38.7 ± 2.9% for *not[1]* alone; Fig. 8 M). Taken together with the data above, we conclude that *ex* is a transcriptional target of *not* that is required for its function in BCs. Previous studies have shown that overexpression of Capping protein B (*cpb[+]*), which antagonizes Enabled by competing for binding F-actin barbed ends and preventing actin polymerization, is capable of complementing impaired Hippo signaling (loss of *warts*) in BCs. Correspondingly, we find that *cpb[+]* is able to significantly

rescue *not[1]*-associated defects in F-actin polarity and collective cell migration (Fig. 8, I–M). Interestingly, we also saw a partial recovery in the Crb distribution in *cpb[+] not[1]* BC clusters (Fig. 8 K), indicating a role for the actin cytoskeleton in controlling Crb polarity.

## Discussion

### A *not*-mediated transcriptional program establishes F-actin polarity during collective migration

We report that *Drosophila* USP22, encoded by *not*, is necessary for F-actin polarity and collective cell migration of invasive BCs

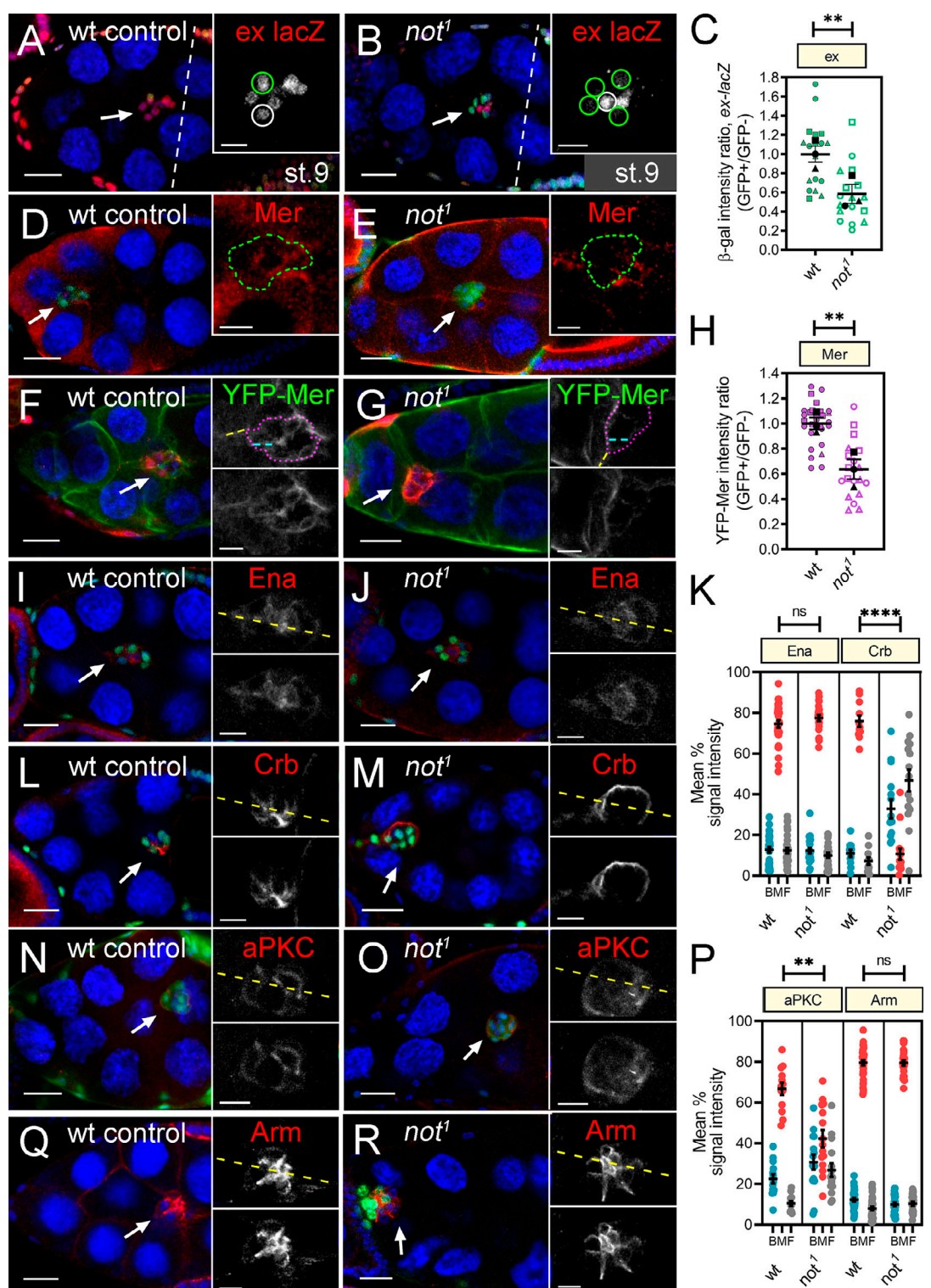

Figure 6. ***not* is required for the normal level and/or distribution of Hippo signaling components and polarity determinants in BCs. (A and B)** Confocal micrographs showing stage 10 egg chambers, except where indicated, with either WT or *not¹* GFP or *lacZ⁺*-labeled clones (green) stained with antibodies against β-gal to detect *ex-lacZ* expression. **(D–R)** Mer (D and E), YFP-Mer (F and G), Ena (I and J), Crb (L and M), aPKC (N and O), or Arm (Q and R) are shown in red, with the exception of YFP-Mer, which is shown in green. Nuclei are stained with TO-PRO-3 (blue). Scale bars are 25 μm (insets, 10 μm). Arrows indicate BCs, insets are magnified images with regions of interest, and yellow lines through clusters mark the position of line scans used for quantitation. **(A)** Mosaic BC clusters, showing the normal expression of *ex-lacZ* in GFP-labeled control clones (green outline) and their siblings (white outline). **(B)** Notably, there is a reduction in *ex-lacZ* expression in *not¹* clones (green outline) compared with control sibling cells (white outline). **(C)** Quantitation of *ex-lacZ* expression; plots show overall mean ± SEM derived from means of *n* ≥ 6 egg chambers/replicate superimposed on plots of the individual measurements. **(D and E)** Mer staining is weak but clearly detectable at the inner-BC junctions in control clones (green outline) but in E, is lost in GFP-labeled *not¹* cells (green outline) and not

adjacent control cells of the same cluster. **(F and G)** A similar effect was observed with YFP-Mer in *lacZ*[+]-labeled clones (β-gal in red). Inset shows regions used to measure the ratio of BC junctional staining inside the cluster (blue dashed line) to nurse cell junctional staining outside the cluster (yellow dashed line). **(H)** Quantitation of YFP-Mer expression (*n* ≥ 6 egg chambers/replicate). **(I and J)** Ena is predominantly located at cell junctions around the polar cells and at inner and outer BC membranes in both control (I) and *not*[1] (J) clones. **(K)** Dot plots of area under curve for back, middle, and front (BMF) of the cluster derived from line scans of signal intensity in Ena and Crb taken from multiple egg chambers, showing mean ± SEM of replicates. **(L and M)** Crb is distributed at inner BC junctions in control BC clusters but in M, is strikingly redistributed at the cortex of *not*[1] BC clusters. **(N and O)** aPKC is distributed at inner BC junctions in control BC clusters, but in O, this distribution is disrupted in *not*[1] clones, with some loss of aPKC at the inner membranes and a more cytoplasmic distribution in the BCs. **(P)** Dot plots of intensity measurements (as in K) for aPKC and Arm. Sample size for dot plots was *n* ≥ 4/replicate. **(Q and R)** The adherens junction protein Arm is apically localized at inner junctions in controls and in *not*[1] BC clusters. **, $P < 0.01$; ****, $P < 0.001$ by unpaired two-tailed Student's *t* test.

(Fig. 9). Collective BC migration requires actomyosin polymerization and contraction at the cortex around the cluster as it moves over the nurse cell substrate; F-actin is effectively excluded from the center of the cluster where polarity determinants acting via the Hippo complex block the activity of the F-actin regulator Enabled. Not has been reported to regulate the actin cytoskeleton directly by promoting the stability of Scar/WAVE. However, we did not observe a reduction in Scar levels in *not* mutant clones, and *scar* loss of function did not disrupt F-actin polarity. Furthermore, we did not observe a significant change in the number of actin protrusions following *not* loss of function, which might be expected if Scar were a target in BCs. Interestingly, *scar* RNAi weakly suppressed *not* loss of function, suggesting that accumulation of branched actin, mediated by Scar at BC–BC junctions, may contribute to disrupted cell polarity and impaired migration. Our data suggest that *not* regulates inside-out F-actin polarity by regulating the expression of Hippo signaling components *ex* and *mer*, which are direct Not targets, in a *yki*-independent manner. Reanalysis of ChIP-Seq data from embryos indicates that Not and Ada2b bind other core Hippo pathway components, so expression of multiple components may be affected by loss of SAGA components. However, *ex* and *mer* are targets for Not, but not Ada2b, which is largely dispensable for migration. Notably, we find that overexpression of *ex* suppressed *not*[1]-induced F-actin accumulation at inner BC junctions, consistent with partial restoration of Hippo function and inhibition of Enabled function. We also observed that *cpb* overexpression rescued loss of *not*, again consistent with disruption of Enabled function due to competitive binding of Cpb to F-actin barbed ends and the inhibition of F-actin polymerization at inner BC junctions. Incomplete rescue of *not*[1] with overexpressed *ex* or *cpb* means that other parallel downstream targets that contribute to *not* function may exist. Interestingly, our data suggest that *not* is dispensable in polar cells for BC migration. It will be interesting to examine whether the requirement for *not* in Hippo pathway function is limited to situations where the Hippo complex acts in a *yki*-independent fashion. The nature of putative noncell autonomous signaling mediated by *not* controlling polar cell number remains to be elucidated, but altered signaling may be an indirect consequence of changes in polarity or via direct changes in the expression of affected signaling molecules.

### *not* regulates the distribution of polarity determinants

A striking effect of *not* loss of function in BCs is the redistribution of Crb from inner to outer BC junctions. When we looked at possible effects of this on other polarity determinants, we

found that localization of aPKC to the inside apical junction between BCs was disrupted, consistent with studies showing that Crb, acting together with the Par complex and endocytic recycling machinery, is necessary for ensuring its correct distribution (Wang et al., 2018). Mislocalized aPKC generates protrusions at the side and back of BCs (Wang et al., 2018), just as we have seen in *not*[1] clusters. Why is Crb mislocalized to the cortex of the BC complex? Our complementation experiments (Fig. 8) suggest that this is partially accounted for by loss of expression of the FERM domain proteins Ex and Mer, which in follicle cells act together with Moe to recruit Crb to the apical surface (Aguilar-Aragon et al., 2020). Moe stabilizes Crb at the apical membrane of epithelia by linking Crb to cortical actin (Médina et al., 2002). Although the physical interaction between Moe and Crb may be weak (Sherrard and Fehon, 2015), Moe is an important regulator of dynamic Crb localization because it acts to antagonize interactions between Crb and aPKC at the marginal zone of the apical membrane domain while stabilizing interactions between Crb and the apical surface (Sherrard and Fehon, 2015). Importantly, in BCs, Moe is cortically localized where it organizes a supercellular actin cytoskeleton network and promotes cortical stiffness (Ramel et al., 2013). An attractive hypothesis, therefore, is that Moe, along with other proteins, is a sink for Crb at the cortex of the BC cluster following loss of Ex and Mer at inner BC junctions in *not* mutants. When we overexpressed *ex*, the normal pattern of Crb localization was partially restored in support of there being competitive binding. Interestingly, we also observed weak rescue of Crb localization following Cpb overexpression. This might be because Moe, or other proteins that tether Crb on the outer membrane, is only accessible in the absence of a strong supercellular F-actin cortex and that restoration of cortical F-actin in *not*[1] *cpb*[+] cells displaces Crb. In WT BCs, Crb needs to be constantly moved from the outside membrane in a dynamin- and Rab5-dependent manner (Wang et al., 2018). Another possibility therefore, which is not mutually exclusive of the first, is that polarization of the F-actin cytoskeleton is important for correct trafficking of Crb in BCs as it is in follicle cells (Aguilar-Aragon et al., 2020).

### SAGA-independent roles for *not* during development and disease

The growth, specification, and migration of cells during tissue development requires precisely regulated patterns of gene expression that depend on numerous cues for temporal and spatial gene activation involving crosstalk with multiple signaling pathways. Strikingly, it has emerged that factors once considered to be ubiquitous regulators of transcription, including the

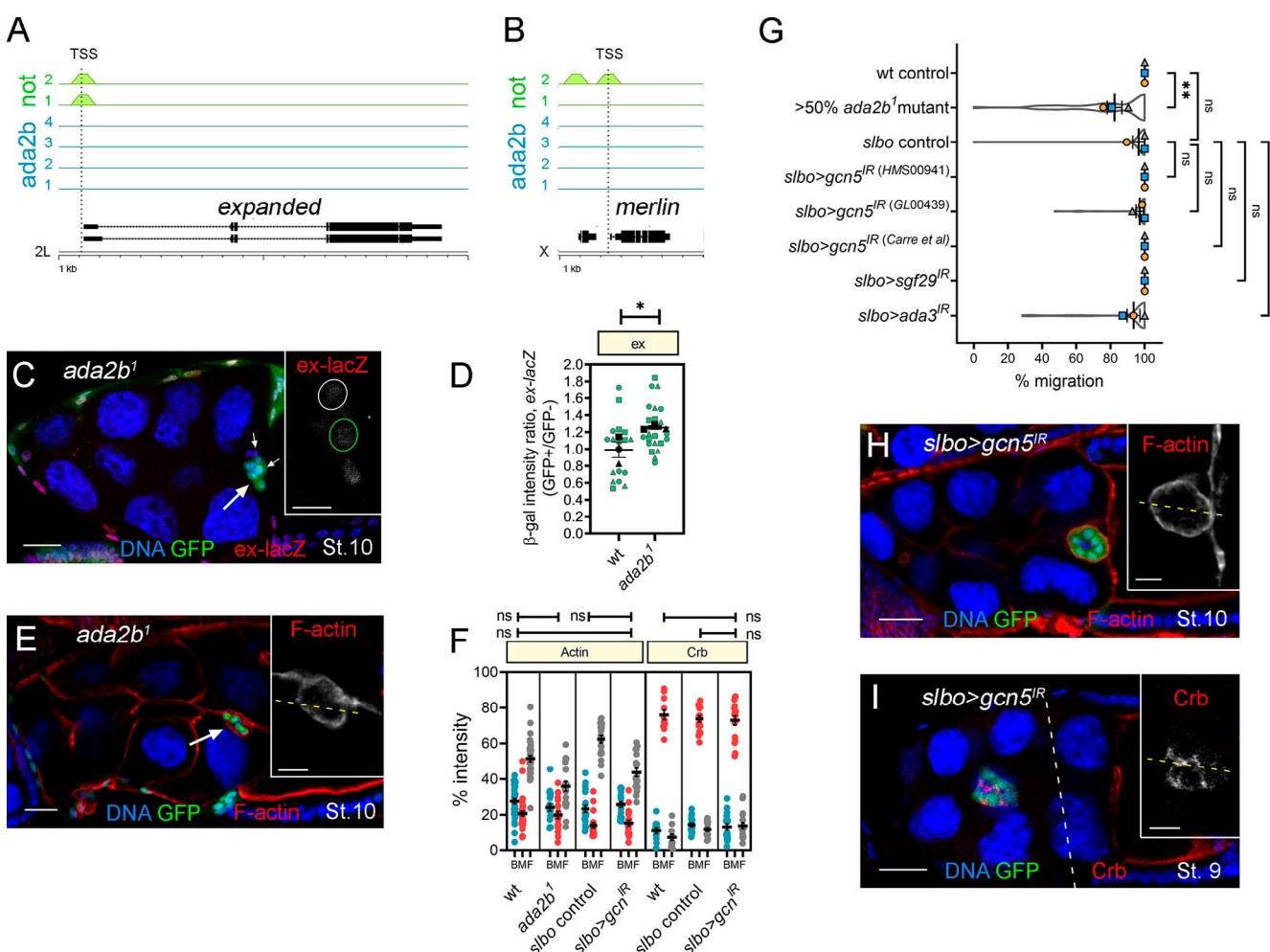

Figure 7. **Ex and Mer are targets of Not but not the HAT module of SAGA, which is dispensable for BC migration. (A and B)** Not, but not Ada2b, binds to the *ex* and *mer* promoters. At the top are the ChIP-binding profiles for all replicates of Not (green; *n* = 2) and Ada2b (blue; *n* = 4) at *ex* and *mer* promoters in *Drosophila* embryos as determined from data reported in Li et al. (2017). Position of the transcription start site (TSS) is shown with a dotted line. Below is a schematic of the gene structure at the respective genomic loci with exons (thick lines) and introns (thin lines). Scale is in 1-kb intervals. **(C)** Confocal micrograph of a stage 10 egg chamber containing a BC cluster (arrow) with *ada2b[1]* GFP-labeled MARCM clones (green) stained with antibodies against β-gal (red) to detect *ex-lacZ* expression. Inset shows *ex-lacZ* staining in grayscale, with mutant cells outlined (green dotted line). **(D)** Beeswarm plot showing a modest increase in *ex-lacZ* staining in *ada2b[1]* mutant cells compared with sibling control cells; mean ± SEM of three repeats (*n* = 6 egg chambers/replicate). **(E)** Confocal micrograph of a stage 10 egg chamber (arrow) with *ada2b[1]* GFP-labeled MARCM clone (green) stained with phalloidin to label F-actin (red), showing F-actin is localized to outer BC junctions as WT (compare with Fig 3 A). **(F)** Dot plots of Actin and Crb intensity for back, middle, and front (BMF) of the cluster derived from line scans taken from several egg chambers (*n* ≥ 3 egg chambers/replicate). **(G)** Plots showing overall mean migration ± SEM derived from means of each replicate (*n* ≥ 15 egg chambers) superimposed on violin plots showing distribution of the migration measurements for each of the indicated genotypes. **(H and I)** Confocal micrographs of egg chambers expressing *slbo>gcn5[IR]* in outer BCs (green) that have been stained for F-actin (H) or Crb (I) in red. Insets show F-actin and Crb staining; dashed yellow line indicates the position of line scans for quantitation (as in F). In all images, nuclei are labeled with TO-PRO-3 (blue). Scale bars are 25 μm (insets, 10 μm). *, P < 0.05; **, P < 0.01 by unpaired two-tailed Student's *t* test. St., stage.

SAGA chromatin-modifying complex, can have specific roles in discrete developmental processes. Although it has been suggested that SAGA is required for all transcribed genes in some contexts (Bonnet et al., 2014), numerous studies have shown that loss of SAGA components affects the expression of only a subset of genes (Pahi et al., 2015; Pankotai et al., 2013; Zsindely et al., 2009) and that different components modulate distinct and overlapping subsets (Helmlinger et al., 2008; Helmlinger et al., 2011; Lee et al., 2000; Weake et al., 2008). These differences in expression are likely to explain their different physiological roles; for instance, during female germline development in *Drosophila*, *ada2B* affects the expression of many genes and is

required for oogenesis, whereas *not* affects relatively few and is dispensable (Li et al., 2017). Genome-wide ChIP studies indicate that even though both DUB and HAT modules bind the same genes, many of the targets do not require the DUB module for expression, explaining the observed dependencies. These experiments also revealed nonoverlapping sites of chromatin occupancy for the DUB and HAT modules of SAGA in *Drosophila* (Li et al., 2017), but the significance of differences in transcriptional targeting for cell function had not been established. Notably, in this respect, we find that the requirement for *not* in BC migration is not matched by a requirement for HAT components, including *ada2b* or *gcn5*. Furthermore, Ada2b has not been found to

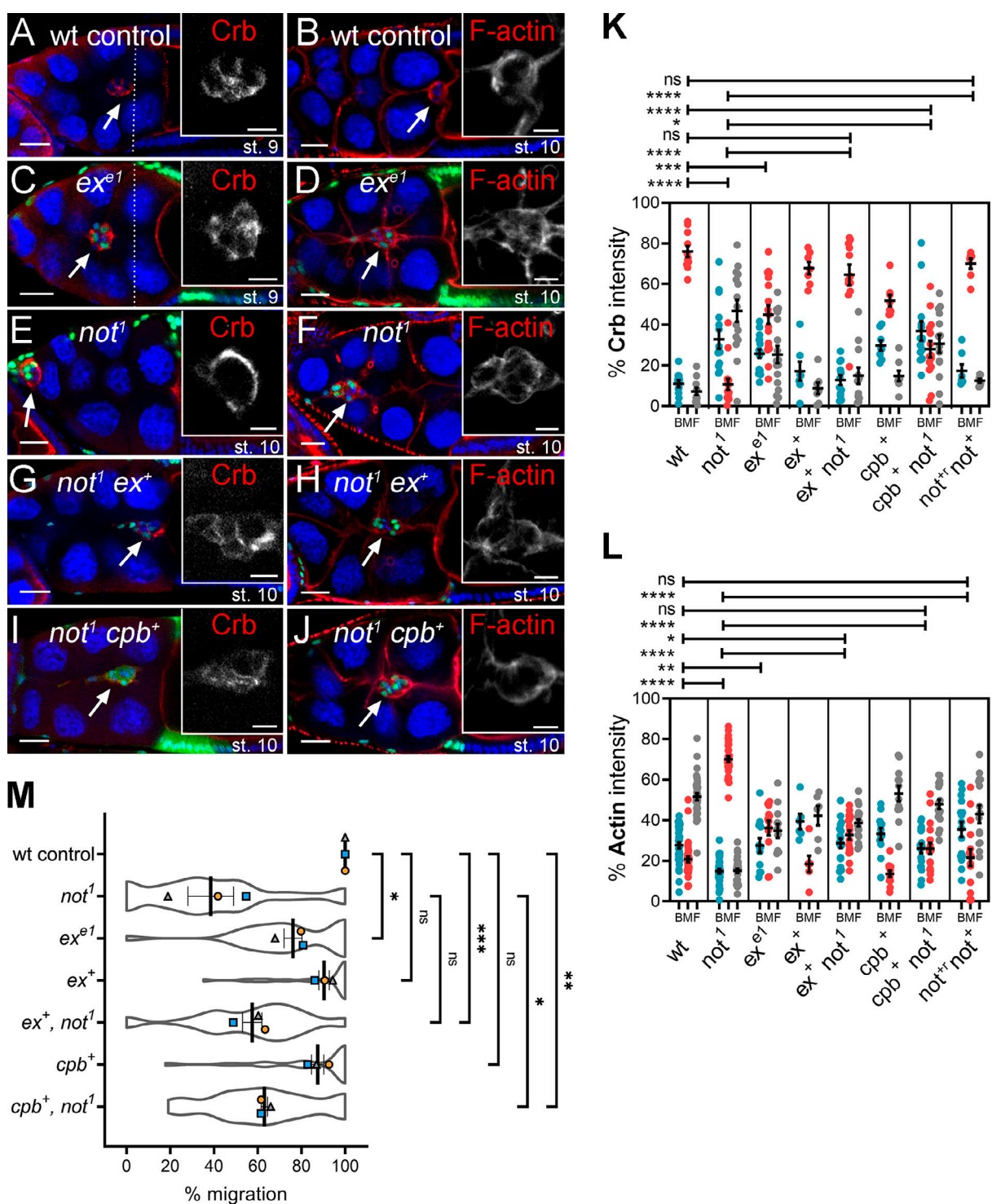

Figure 8. **Overexpression of *ex* or the actin capping protein *cpb* partially rescues cell migration and polarity defects. (A–J)** Confocal micrographs of Crb or F-actin (red) staining in egg chambers harboring GFP-labeled MARCM clones (green) of different genotypes. TO-PRO-3 (blue) labels all nuclei. Scale bars are 25 µm (insets, 10 µm). The stage of egg chamber development is as indicated, with dotted lines showing the position of overlying centripetal follicle cells in stage 9 chambers. BCs are indicated with arrows. **(A)** Control showing normal distribution of Crb at contacts between the BCs inside the cluster. **(B)** Control showing normal cortical distribution of F-actin around the outer membrane of the cluster. **(C)** *ex^e1* clones showing partial disruption of Crb. **(D)** F-actin polarization is also partially impaired in *ex^e1* BCs, with some F-actin visible at inner junctions of the migrating clusters. **(E)** Crb is redistributed away from inner junctions to the cortex of the cluster in *not^1* clones. **(F)** F-actin is found distributed on inner junctions of *not^1* clusters between BCs. **(G)** The disruption of Crb localization in *not^1* clones is partially rescued by overexpression of *ex* (*not^1 ex^+*). **(H)** F-actin also is more normally polarized in *not^1 ex^+* BCs, although some weak staining is also evident between BC–BC junctions. **(I)** Overexpression of cpb weakly restores some Crb distribution in *not^1* cells (*not^1 cpb^+*). **(J)** F-actin is displaced from BC junctions inside the *not^1 cpb^+* clusters. **(K)** Quantification of mean percentage of Crb staining at the back, middle, and front (BMF) of the cluster (area under curve measurements) derived from *n* > 5 egg chambers. Results of two-way ANOVA comparisons of mean ratio of Crb staining in the middle

of the cluster are shown above. **(L)** Quantification of mean percentage of F-actin staining at the BMF of the cluster (area under curve measurements) derived from n > 6 egg chambers. Two-way ANOVA comparisons of mean ratio of F-actin staining in the middle of the cluster are also shown. **(M)** Plots showing overall mean migration ± SEM derived from means of each replicate (n ≥ 15 egg chambers) of the indicated genotypes. *, P < 0.05; **, P < 0.01; ***, P < 0.001; ****, P < 0.0001 by unpaired two-tailed Student's t test. st., stage.

bind the *ex* and *mer* promoters, providing a molecular explanation for *not*'s SAGA-independent role. Importantly, these findings challenge the perceived view that transcriptional roles for not/USP22 are mediated solely by SAGA. This may have broader relevance to situations where USP22, but not other members of SAGA, is associated with human disease states, particularly where cell polarity is frequently disrupted, such as cancer (Glinsky et al., 2005). Our current efforts are directed at identifying SAGA-independent factors that facilitate Not's chromatin binding and function.

## Materials and methods

### *not* transgene
An RNAi-resistant, full-length *not* expression construct was synthesized by GeneArt (Invitrogen). RNAi resistance was achieved by incorporating numerous silent polymorphic mutations such that in the regions targeted by dsRNAs, homology with the inverted repeat sequences was limited to no more than eight contiguous base pairs (Jonchere and Bennett, 2013). The *not* open reading frame was shuttled into pPMW-attB (Chen et al., 2015) by gateway cloning, placing the *not* open-reading frame downstream of a Myc epitope tag. Stable transgenic flies were made by phiC31 integrase-mediated transgenesis at a landing site on the second (attP40 at 25C6) and third (attP2 at 68A4) chromosomes by the University of Cambridge Fly Facility.

### *Drosophila* stocks and genetics
Flies were raised and crossed at 25°C according to standard procedures. w[1118] or flippase recognition target (FRT) 80B flies were used as the WT control strains. 133 RNAi lines, corresponding to 45 *Drosophila* DUBs (Table S2), were initially screened for BC defects by crossing to *UAS-Dcr2; slbo-GAL4, UAS-GFP/CyO* with confirmatory crosses (in the absence of Dcr2) to *slbo-GAL4, UAS-GFP, His2Av-mRFP/CyO* and *c306-GAL4; slbo-GAL4, UAS-GFP, UAS-DsRed[NLS]/CyO*. *UAS-not[IR]* (#45776; Vienna Drosophila Resource Center [VDRC]) was identified as having the most severe effect on migration. The FLP/FRT site-specific recombination system was used to generate mutant clones with a heat shock promoter (Xu and Rubin, 1993). The following fly lines were obtained from the Bloomington Drosophila Stock Center unless otherwise noted: *FRT80B* (BL1988); *w[1118]* (BL6409); *slbo-Gal4, UAS-GFP* (BL6458, Montell Lab); *slbo-lacZ* enhancer trap line (BL12227); *slbo-Lifeact-GFP* (BL58364); *c306-Gal4, UAS-GFP* (BL3743); *upd-Gal4, UAS-cas9.P2* (BL58985); *UAS-not[sgRNA]* (BL84220, M{WKO.P5-C6}); *tubGal80 FRT80B* (BL5191); *A>γ>Gal4, UAS-lacZ* (BL4410); *UAS-Scar[IR]* (21908; VDRC); *UAS-gcn5[IR]* (BL9332, BL35601, BL33981); *UAS-sgf29[IR]* (BL39000); *UAS-ada3[IR]* (46320; VDRC); and *UAS-yki[IR]* (104523; VDRC).

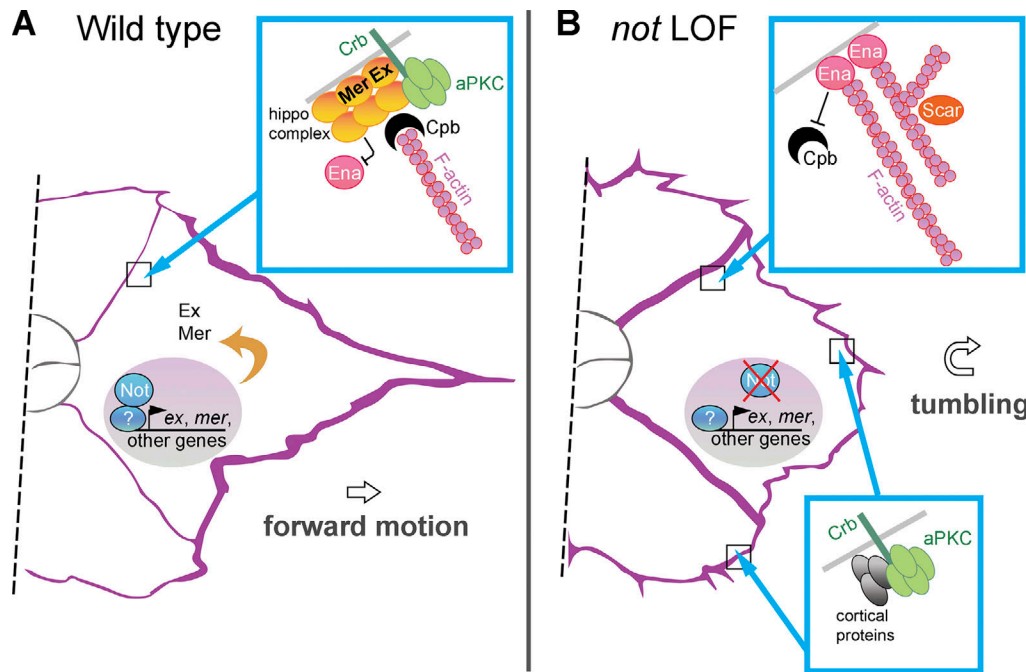

Figure 9. **Schematic illustration. (A)** Depiction of *not*'s role in BC migration. **(B)** Depiction of the effect of *not* loss of function (LOF) on protein complexes described in the text. In conjunction with other factors, Not associates with, and regulates the expression of, *ex* and *mer*, which function to anchor the Hippo complex, including Crb, to the BC–BC junction, suppressing Ena-driven F-actin (purple) formation there. The expression of other genes is also likely to be affected. Crb is displaced in *not* LOF, and F-actin accumulates at the BC–BC junction, leading to tumbling of the cluster.

For clonal analysis, we used the following strains: *hsFLP, tub-Gal4, UAS-GFP; +/+; tubGAL80 FRT80B/TM6B* (generated from BL42732, BL5191); *hsFLP, tub-Gal4, UAS-GFP; tubGAL80 FRT40A/+; +/TM6B* (generated from BL42732, BL5192); *hsFLP, tub-Gal4, UAS-GFP; +/+; FRT82B tubGAL80/TM6B* (generated from BL42732, BL44408); and *hsFLP; A>y>Gal4, UAS-lacZ; tubGal80 FRT80B/ SM5-TM6B* (generated from BL4410, BL5191).

The amorphic *not* allele *not[1]* was obtained from Margarete Heck (University of Edinburgh, Edinburgh, UK) and recombined with *FRT80B. FRT82B ada2b[1]* was a gift from Jerry Workman (Stowers Institute for Medical Research, Kansas City, MO; Li et al., 2017). *UAS-Scar* and *FRT40A Scar[Δ37]* were gifts from Eyal Schejter (Weizmann Institute of Science, Rehovot, Israel). *UAS-cpb, UAS-ex* (Lucas et al., 2013), *upd-lacZ* (Jiang et al., 2009), and *ex-lacZ* (Fletcher et al., 2012) were gifts from Nic Tapon (The Francis Crick Institute, London, UK). *Mer-YFP* (Su et al., 2017) was a gift Rick Fehon (The University of Chicago, Chicago, IL). Information on these strains is also available at http://www.flybase.org.

## Generation of mosaic clones using mosaic analysis with a repressible cell marker
Mosaic analysis with a repressible cell marker (MARCM) was used to generate positively marked clones labeled with GFP (Lee and Luo, 2001). Expression of genes under GAL4-UAS is inhibited in the presence of GAL80. Heat shocking induces the expression of heat shock–driven FLP, which acts to induce recombination at FRT. Homozygous daughter cells lacking GAL80 are then capable of GAL4-mediated gene expression of GFP and other UAS transgenes. Mitotic recombination is initiated after heat shock, where some daughter cells are GFP+ while others are GFP− due to the presence of GAL80. To obtain BC mitotic (mosaic) clones, progeny of the right genotypes were heat shocked twice a day for 1 h each with at least 5-h intervals between treatments from pupae to adult at 37°C. Newly enclosed adults (2–3 d old) were fattened for 2 d on yeast paste.

## Immunofluorescent staining
Ovaries were dissected in PBS and fixed with 3.7% PFA in PBS. The ovaries were washed with PBS plus Tween 20 (PBST; 1× PBS, 0.2% Tween 20) three times for 15 min each time. Ovaries were then blocked with PBST and FBS (PBTB; 1× PBS, 0.2% Tween 20, 5% FBS) for 1 h at room temperature. The ovaries were treated with primary antibodies in PBTB at 4°C overnight. The following primary antibodies from the Developmental Studies Hybridoma Bank were used for immunostaining: mouse anti-Armadillo (N27A1, 1:200, concentrate), mouse anti-Enabled (5G2, 1:25, concentrate), mouse anti-β-gal (40-1a, 1:300, concentrate), mouse anti-Fasciclin 3 (7G10, 1:200, supernatant), mouse anti-Eyes Absent (eya10H6, 1:100, supernatant), and mouse anti-SCAR (P1C1, 1:200, concentrate). Mouse anti-aPKC-ζ (sc-17781, 1:200) was from Santa Cruz Biotechnology. Guinea pig anti-Merlin (1:7,500) was from the R. Fehon laboratory, The University of Chicago. Chicken anti-Moesin (1:200) was from Gregory Emery (University of Montreal, Montreal, Quebec, Canada; Plutoni et al., 2019). The primary antibodies were washed with PBST three times for 15 min and then blocked with

PBTB for 1 h at room temperature. Ovaries were incubated with Alexa Fluor–conjugated secondary antibodies (1:500, Life Technologies) in PBTB at 4°C overnight. Phalloidin 555 (1:50; Molecular Probes) was used to stain F-actin. Ovaries were washed with PBST for 15 min before staining nuclei with TO-PRO-3 (1:1,000; Life Technologies) in PBST for 15 min. Ovaries were mounted in VECTASHIELD (Vector Laboratories). For Crb staining, ovaries were dissected in PBS and fixed with boiled 8% PFA in PBS and heptane (6:1) for 10 min. Samples were treated with heptane and methanol (1:2) for 30 s. They were then washed in methanol for 10 min. The ovaries were washed with PBST two times for 15 min each time. Ovaries were then blocked with PBTB for 30 min at room temperature. The ovaries were treated with mouse anti-Crumbs (Cq4, 1:100, concentrate; Developmental Studies Hybridoma Bank) in PBTB at 4°C overnight.

## Image acquisition and analysis of fixed samples
Images were taken on a confocal microscope (LSM 710 or LSM 780; Carl Zeiss) using 20×/0.5 NA air objectives. Three laser lines were used based on the excitation of wavelength of the staining dyes, which included 488-nm, 561-nm, and 633-nm wavelengths. Extent of migration (the migration index) was measured as a percentage of the distance traveled to the oocyte/nurse cell boundary in stage 10 egg chambers. ImageJ software (https://imagej.nih.gov/ij/) was used for quantification of signal intensities in mosaic clusters using z-stack maximum projections. Raw integrated density was used as the intensity value. For line scan profiles, maximum intensity images of Actin, Crb, Scar, Enabled, aPKC, Arm, and Moe staining were generated in ImageJ. Background signals were subtracted. The plot profile function in ImageJ was used to measure signal intensities along lines drawn through the center of BC clusters, and the peak analyzer tool in OriginPro (OriginLab) was used to calculate the area under peaks that were identified. The ratio of intensities at front, middle, and back were compared and normalized in GraphPad Prism 8 (GraphPad Software). Experiments were performed in triplicate; the number of egg chambers dissected is reported in the figure legends. The following statistical tests were performed using GraphPad Prism 8: Student's *t* tests, one-way or two-way ANOVA with Tukey correction for multiple comparisons, and multiple linear regression with least squares. Data distribution was assumed to be normal, but this was not formally tested in every case. Figures were made using FigureApp in OMERO (Allan et al., 2012; Burel et al., 2015) and final assembly in Adobe Photoshop.

## Egg chamber culture and time-lapse imaging of live egg chambers
Live imaging of egg chamber culture was as previously described (Law et al., 2013; Prasad et al., 2007), with slight modification. Briefly, media for both dissection and live imaging comprising Schneider media (Gibco), 15% FBS, 0.1 mg/ml acidified insulin (Sigma), 9 μM FM4-64 dye (Molecular Probes), and 0.1 mg/ml penicillin-streptomycin (Gibco) were freshly prepared. The pH of the media was adjusted to 6.90–6.95. Individual egg chambers from well-fattened progeny of the right genotype were dissected and transferred to borosilicate glass bottom

chambered coverglasses (Thermo Fisher Scientific) for imaging. Imaging was done at 25°C. Time-lapse movies were acquired on an inverted confocal microscope (LSM 710; Carl Zeiss) using 20×/0.5 NA air objectives. Two laser lines were used based on the excitation of wavelength of the endogenous GFP and FM4-64 dye, which are 488-nm and 561-nm wavelengths, respectively. 16–20 slices of z-stacks were taken with 2.5-μm slices every 3 min.

## Analysis of time-lapse images

Time-lapse image analyses were performed using a custom macro for ImageJ to analyze the behavior of BC migration and extension dynamics (Law et al., 2013; Poukkula et al., 2011), with slight modification. Briefly, time-lapse movies were split into different channels. Maximum projections of the GFP channel were created. Egg chambers were rotated so that anterior ends were at the left. BCs were manually thresholded to mask nuclear GFP generated from the MARCM system through the first or early phase of migration. Images of BC clusters were then segmented into cell body and cellular extensions using signals from *slbo-LifeAct-GFP*. Extensions were grouped based on their positions in relation to the leading edge of the cluster: front (315–45°), side (45–135° or 225–315°), and back (135–225°). The macro also enabled tracking of the movement of the cluster to measure the migration speed. Forward-directed speed was calculated on the x axis by taking the distance of the center of the cluster at one time point relative to the next time point. The tumbling index was calculated as the mean percentage of frames per time-lapse movie that showed rounded clusters, exhibiting changes in the position of individual cells within the cluster for two or more consecutive frames in the first half of migration. Data were collated in Microsoft Excel, and independent Student's *t* tests were done with GraphPad Prism 8. For visualization of stills (Fig. 4, A, A', B, and B'), GFP-labeled nuclei were segmented in Imaris (Bitplane) and labeled in white.

## Analysis of previously reported ChIP datasets

ChIP-seq data were downloaded from Gene Expression Omnibus under accession no. GSE98862; the dm3 assembly of the *Drosophila* genome was obtained from the University of California, Santa Cruz (http://www.genome.ucsc.edu/cgi-bin/hgTables). Peaks from Ada2b and Not ChIP experiments were mapped to the dm3 genome assembly using BEDTools software (Quinlan and Hall, 2010), and any genes matching to peaks from −1,000 to +200 of the transcription start site were identified. For visualization of ChIP-seq peaks on the genome, we used the 'karyoploteR' R/Bioconductor package (Gel and Serra, 2017).

## Genotypes of strains

Fig. 1 A, w[1118]/+; slbo-Gal4, UAS-GFP/+; Fig. 1 B, slbo-Gal4, UAS-GFP/UAS-not[IR]; Fig. 1 C, slbo-Gal4, UAS-GFP; UAS-not[+r]/+; Fig. 1 E, (as A–C) with slbo-Gal4, UAS-GFP/UAS-not[IR]; UAS-GFP/+ and slbo-Gal4, UAS-GFP/UAS-not[IR]; UAS-not[+r]/+; Fig. 1 F, hsFLP, tub-Gal4, UAS-GFP/ upd-lacZ ;; +, FRT80B/tub-Gal80, FRT80B; Fig. 1 G, hsFLP, tub-Gal4, UAS-GFP/ upd-lacZ ;; not[1], FRT80B/tub-Gal80, FRT80B; Fig. 1 H, (as E and F) with hsFLP, tub-Gal4, UAS-GFP/+ ; UAS-not[+r]/+ ; not[1], FRT80B/tub-Gal80, FRT80B;

Fig. 1 I, hsFLP, tub-Gal4, UAS-GFP/+ ;; not[1], FRT80B/tub-Gal80, FRT80B; Fig. 1 J, (as H) with hsFLP, tub-Gal4, UAS-GFP/ upd-lacZ ;; +, FRT80B/tub-Gal80, FRT80B and hsFLP, tub-Gal4, UAS-GFP/+ ; UAS-not[+r]/+ ; not[1], FRT80B/tub-Gal80, FRT80B.

Fig. 2 A, hsFLP, tub-Gal4, UAS-GFP/upd-lacZ ;; not[1], FRT80B/tub-Gal80, FRT80B; Fig. 2 B, hsFLP, tub-Gal4, UAS-GFP/upd-lacZ ;; +, FRT80B/tub-Gal80, FRT80B and hsFLP, tub-Gal4, UAS-GFP/upd-lacZ ;; not[1], FRT80B/tub-Gal80, FRT80B; Fig. 2 C, hsFLP, tub-Gal4, UAS-GFP/+ ;; not[1], FRT80B/tub-Gal80, FRT80B; Fig. 2 D, hsFLP, tub-Gal4, UAS-GFP/upd-lacZ ;; not[1], FRT80B/tub-Gal80, FRT80B; Fig. 2 E, hsFLP, tub-Gal4, UAS-GFP/+ ;; not[1], FRT80B/tub-Gal80, FRT80B; Fig. 2 F, hsFLP, tub-Gal4, UAS-GFP/+ ;; not[1], FRT80B/tub-Gal80, FRT80B; Fig. 2 G, hsFLP, tub-Gal4, UAS-GFP/+; slbo-lacZ/+; +, FRT80B/tub-Gal80, FRT80B; Fig. 2 H, hsFLP, tub-Gal4, UAS-GFP/+; slbo-lacZ/+; not[1], FRT80B/tub-Gal80, FRT80B; Fig. 2 I, Quantification of G,H; Fig. 2 J, hsFLP, tub-Gal4, UAS-GFP/+ ;; +, FRT80B/tub-Gal80, FRT80B; Fig. 2 K, hsFLP, tub-Gal4, UAS-GFP/+ ;; not[1], FRT80B/tub-Gal80, FRT80B; WT GFP[−]: hsFLP, tub-Gal4, UAS-GFP/+; slbo-lacZ/+; +, FRT80B/tub-Gal80, FRT80B (or homozygous for tub-Gal80, FRT80B); WT GFP[+]: hsFLP, tub-Gal4, UAS-GFP/+; slbo-lacZ/+; +, FRT80B/+, FRT80B; not[1] GFP[−]: hsFLP, tub-Gal4, UAS-GFP/+; slbo-lacZ/+; not[1], FRT80B/tub-Gal80, FRT80B (or homozygous for tub-Gal80, FRT80B); not[1] GFP[+]: hsFLP, tub-Gal4, UAS-GFP/+; slbo-lacZ/+; not[1], FRT80B/ not[1], FRT80B.

Fig. 3, WT control: hsFLP, tub-Gal4, UAS-GFP/+ ;; +, FRT80B/tub-Gal80, FRT80B; not[1]: hsFLP, tub-Gal4, UAS-GFP/+ ;; not[1], FRT80B/tub-Gal80, FRT80B; not[1]; tub>not[+r]: hsFLP, tub-Gal4, UAS-GFP/+ ; UAS-not[+r]/+ ; not[1], FRT80B/tub-Gal80, FRT80B.

Fig. 4, Control: hsFLP, tub-Gal4, UAS-GFP/+; slbo-LifeAct-GFP/+; +, FRT80B/tub-Gal80, FRT80B; not[1]: hsFLP, tub-Gal4, UAS-GFP/+; slbo-LifeAct-GFP/+; not[1], FRT80B/tub-Gal80, FRT80B.

Fig. 5 A, A', and A'',. hsFLP, tub-Gal4, UAS-GFP/+ ;; +, FRT80B/tub-Gal80, FRT80B; Fig. 5 B, B', and B'',. hsFLP, tub-Gal4, UAS-GFP/+ ;; not[1], FRT80B/tub-Gal80, FRT80B; Fig. 5, C and H, slbo-Gal4, UAS-GFP/UAS-Scar[wt]; Fig. 5, I and J, hsFLP; tubGAL80, FRT40A/Scar[Δ37], FRT40A; Act>CD2>Gal4, UAS-GFP/+; Fig. 5 K, hsFLP, tub-Gal4, UAS-GFP/+ ; UAS-Scar[IR]/+; not[1], FRT80B/tub-Gal80, FRT80B.

Fig. 6 A, (WT control): hsFLP, tub-Gal4, UAS-GFP/+ ; ex-lacZ/+; +, FRT80B/tub-Gal80, FRT80B; Fig. 6 B, (not[1]): hsFLP, tub-Gal4, UAS-GFP/+ ; ex-lacZ/+; not[1], FRT80B/tub-Gal80, FRT80B; Fig. 6, D, I, L, N, and Q, (WT control): hsFLP, tub-Gal4, UAS-GFP/+ ;; +, FRT80B/tub-Gal80, FRT80B; Fig. 6, E, J, M, O, and R. (not[1]): hsFLP, tub-Gal4, UAS-GFP/+ ;; not[1], FRT80B/tub-Gal80, FRT80B; Fig. 6 F, (WT control): hsFLP/+ ; Mer-YFP/Act>y>Gal4, UAS-lacZ; +, FRT80B/tub-Gal80, FRT80B; Fig. 6 G, (not[1]): hsFLP/+ ; Mer-YFP/Act>y>Gal4, UAS-lacZ; not[1], FRT80B/tub-Gal80, FRT80B.

Fig. 7 C, hsFLP, tub-Gal4, UAS-GFP/+ ; ex-lacZ/+; ada2b[1], FRT82B/tub-Gal80, FRT82B; Fig. 7 D, (WT) hsFLP, tub-Gal4, UAS-GFP/+ ; ex-lacZ/+; FRT82B/tub-Gal80, FRT82B; (ada2b[1]) hsFLP, tub-Gal4, UAS-GFP/+ ; ex-lacZ/+; ada2b[1], FRT82B/tub-Gal80, FRT82B; Fig. 7 E, hsFLP, tub-Gal4, UAS-GFP/+ ;; ada2b[1], FRT82B/tub-Gal80, FRT82B; Fig. 7 F, hsFLP, tub-Gal4, UAS-GFP/+ ;; FRT82B/tub-Gal80, FRT82B; hsFLP, tub-Gal4, UAS-GFP/+ ;; ada2b[1], FRT82B/tub-Gal80, FRT82B; slbo-Gal4, UAS-GFP; slbo-Gal4, UAS-GFP/UAS-gcn5[IR]; Fig. 7 G, hsFLP, tub-Gal4, UAS-GFP/+ ;;

FRT82B/tub-Gal80, FRT82B; hsFLP, tub-Gal4, UAS-GFP/+ ;; ada2b[1], FRT82B/tub-Gal80, FRT82B; slbo-Gal4, UAS-GFP; slbo-Gal4, UAS-GFP/UAS RNAi lines as indicated; Fig. 7, H and I, slbo-Gal4, UAS-GFP/UAS-gcn5[IR] (Carré et al., 2005).

Fig. 8, A and B, hsFLP, tub-Gal4, UAS-GFP/+; ; +, FRT80B/tub-Gal80, FRT80B; Fig. 8, C and D, hsFLP; ex[e1], FRT40A/tub-Gal80, FRT40A; Act>CD2>Gal4, UAS-GFP/+; Fig. 8, E and F, hsFLP, tub-Gal4, UAS-GFP/+; ; not[1], FRT80B/tub-Gal80, FRT80B; Fig. 8, G and H, hsFLP, tub-Gal4, UAS-GFP/+; UAS-ex[+]/+; not[1], FRT80B/tub-Gal80, FRT80B; Fig 8, I and J, hsFLP, tub-Gal4, UAS-GFP/+; UAS-cpb[+]/+; not[1], FRT80B/tub-Gal80, FRT80B; Fig. 8, K–M, (quantitation of A–J together with the following genotypes); hsFLP, tub-Gal4, UAS-GFP/+; UAS-ex[+]/+; +, FRT80B/tub-Gal80, FRT80B; hsFLP, tub-Gal4, UAS-GFP/+; UAS-cpb[+]/+; +, FRT80B/tub-Gal80, FRT80B; hsFLP, tub-Gal4, UAS-GFP/+ ; UAS-not[+r]/+ ; not[1], FRT80B/tub-Gal80, FRT80B.

Fig. S1, A, B, and D, c306-Gal4/+; UAS-GFP; c306-Gal4/+; UAS-GFP/UAS-cas9; c306-Gal4/+; UAS-GFP/UAS-cas9; not[sgRNA]/+; +/+; slbo-Gal4, UAS-GFP; +/+; slbo-Gal4, UAS-GFP/UAS-cas9; not[sgRNA]/+; upd-Gal4/+; UAS-mCherry; upd-Gal4/+; UAS-mCherry/UAS-cas9; not[sgRNA]/+; Fig. S1 C, c306-Gal4/+; UAS-GFP/UAS-cas9; c306-Gal4/+; UAS-GFP/UAS-cas9; not[sgRNA]/+.

Fig. S2, WT control: hsFLP, tub-Gal4, UAS-GFP/+ ;; +, FRT80B/tub-Gal80, FRT80B; not[1]: hsFLP, tub-Gal4, UAS-GFP/+ ;; not[1], FRT80B/tub-Gal80, FRT80B.

Fig. S3 A, w[1118]/+; slbo-Gal4, UAS-GFP/+; Fig. S3 B, slbo-Gal4, UAS-GFP/UAS-yki[IR].

### Online supplemental material

Fig. S1 shows the effects of targeting *not* in the BC cluster with CRISPR-Cas9. Fig. S2 demonstrates that the level and distribution of Moe is unaffected by *not*. Fig. S3 shows that *yki* is dispensable in outer BCs for migration. Video 1 shows a representative time-lapse of normal BC migration. Video 2 shows the effect of *not* RNAi on BC migration. Video 3 shows actin dynamics during normal migration. Video 4 shows effects of *not* loss of function on actin protrusions and BC migration. Video 5 shows early tumbling of a *not* BC cluster that delaminates from the epithelium. Table S1 provides a summary of ChIP-Seq data showing the association of Not and Ada2b with the promoters of Hippo pathway genes. Reported here is the number of times promoter binding was identified by Lin et al. (2014) using ChIP-Seq. Table S2 provides details of RNAi lines used to screen for *Drosophila* DUBs involved in BC migration.

## Acknowledgments

We thank Greg Emery, Rick Fehon, Margarete Heck, Timothy Megraw, Eyal Schejter, Nic Tapon, Jerry Workman, the Developmental Studies Hybridoma Bank, and Bloomington Stock Center for antibodies, vectors, and fly stocks. Thanks also to the Liverpool Computational Biology Facility and Chris Seidel (Stowers Institute) for assistance with ChIP data analysis and to the Liverpool Centre for Cell Imaging for help with microscopy and image analysis.

The work was funded by the Medical Research Council (MR/K015931/1), North West Cancer Research Fund (CR847), Liverpool Cancer Research UK Centre, and the University of Liverpool international PhD fees waiver scheme.

The authors declare no competing financial interests.

Author contributions: H. Badmos: investigation, methodology, validation, formal analysis, conceptualization, visualization, writing — review and editing. N. Cobbe: investigation, methodology, writing — review and editing. A. Campbell: investigation, validation. R. Jackson: formal analysis. D. Bennett: conceptualization, funding acquisition, project administration, supervision, methodology, formal analysis, visualization, writing — original draft, writing — review and editing.

Submitted: 2 July 2020

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

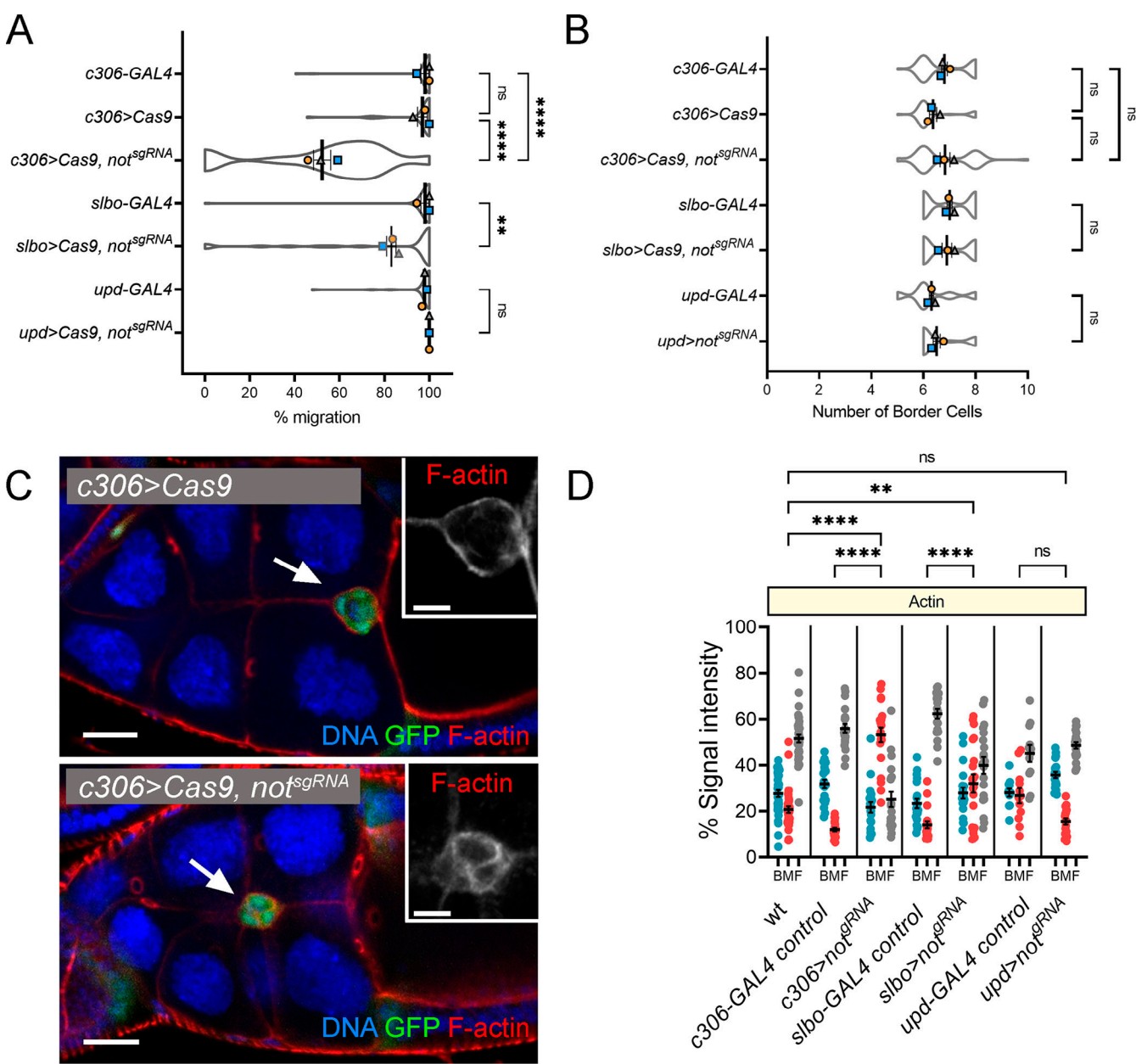

Figure S1. **Effects of targeting *not* in the BC cluster with CRISPR-Cas9. (A)** SuperPlot showing mean migration ± SEM derived from means of each replicate superimposed on violin plots of migration measurements for each of the indicated genotypes. Replicate trials consisted of *n* ≥ 12 egg chambers for each genotype. **(B)** Plots showing mean number of BCs ± SEM derived from means of each replicate superimposed on violin plots of measurements for each of the indicated genotypes. **(C)** Confocal micrographs of egg chambers with the BC cluster labeled with GFP in green (arrow) under the control of *c306-GAL4*. F-actin is in red; TO-PRO-3 (blue) stains all nuclei. A retarded BC cluster (arrow) upon CRISPR-Cas9 targeting of *not* (*c306>Cas9, not^sgRNA*) shows disrupted F-actin polarity (see also insets). Scale bars in confocal images are 25 μm (insets, 10 μm). **(D)** Dot plot quantification of area under curve for back, middle, and front (BMF) of the BC cluster derived from line scans taken through the cluster. Error bars indicate SEM of replicates. **, P < 0.01; ****, P < 0.0001 by one-way ANOVA.

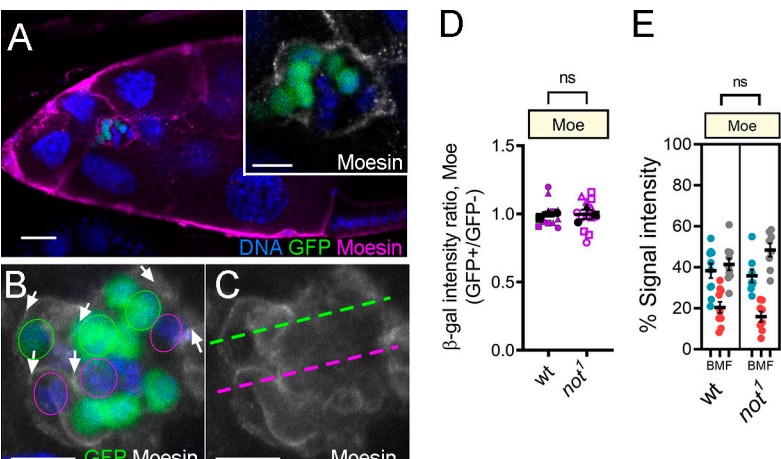

Figure S2. **The level and distribution of Moe is unaffected by *not*.** **(A–C)** Confocal micrographs showing egg chambers with either WT or *not¹* GFP-labeled MARCM clones (green) stained with antibodies against Moe (magenta). DNA stained with TO-PRO-3 is in blue. **(A)** Stage 10 egg chamber; inset shows magnified image of the mosaic BC cluster, with Moe staining in grayscale visible around the periphery of the cluster surrounding both WT and mutant cells. Scale bar is 25 μm (inset, 10 μm). **(B and C)** Another example of a mosaic *not¹* BC cluster to illustrate regions selected for quantitation of Moe staining. **(B)** Regions of outer BC–nurse cell junctions (arrows) were selected adjacent to *not¹* cells (green outline) or WT cells (purple outline). **(C)** Example of positioning of line scans through mutant (green line) and WT (purple line) regions of a mosaic cluster. **(D)** Quantitation of Moe intensity in WT control and *not¹* clones (as in B) compared with control sibling cells. Error bars indicate SEM in three biological replicates. **(E)** Dot plots showing quantification of area under curve for back, middle, and front (BMF) of the BC clusters derived from line scans taken through the cluster (e.g., as in C). Error bars indicate SEM of replicates (*n* ≥ 8). Significance determined by unpaired two-tailed Student's *t* test.

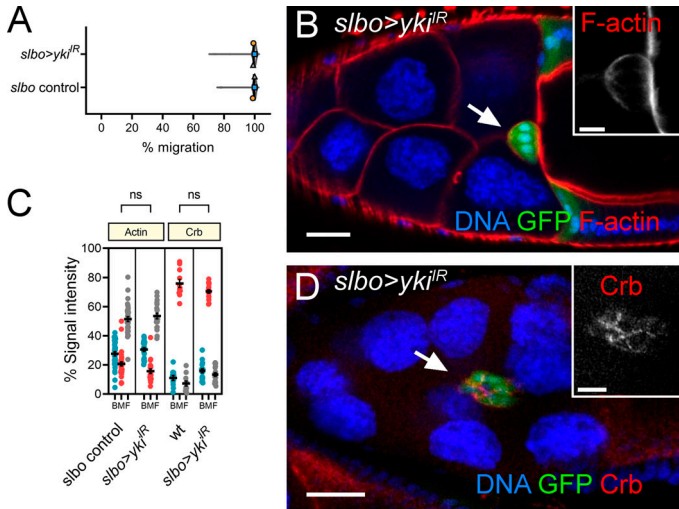

Figure S3. ***yki* is dispensable in outer BCs for migration. (A)** Graph showing mean migration ± SEM derived from means of each replicate superimposed on violin plots of migration measurements for each of the indicated genotypes. Replicate trials consisted of *n* ≥ 20 egg chambers for each genotype. **(B)** Confocal micrograph of an egg chamber with outer BCs coexpressing GFP (green) under the control of *slbo-GAL4*, together with dsRNA for *yki*. F-actin labeled by phalloidin in red and DNA with TO-PRO-3 in blue. **(C)** Dot plots showing quantification of area under curve for F-actin and Crb staining at the back, middle, and front (BMF) of the BC cluster derived from line scans taken through clusters. Error bars indicate SEM of replicates (*n* = 20 clusters). **(D)** Confocal image (as in B) but stained with Crb instead of F-actin. Significance determined by unpaired two-tailed Student's *t* test.

Video 1. **Time-lapse movie of BC migration (10 frames/s) starting from specification of the cluster and the ability of the cluster to acquire forward protrusion, followed by cell-on-cell migration to the anterior border of the oocyte.** GFP expression is driven by *slbo-Gal4* to label the BC cluster in green. Nuclei are labeled with *Ub-His2A-RFP* in magenta. Egg chamber genotype: *w¹¹¹⁸/+; slbo-Gal4, UAS-GFP, Ub-His2A-RFP/+*.

**Video 2.** **Time-lapse movie of abnormal BC migration (10 frames/s) with failure of cluster to detach from the epithelium after RNAi knockdown of _not_ under the control of _slbo-GAL4._** GFP expression is driven by _slbo-Gal4_ to label the BC cluster in green. Egg chamber genotype: _slbo-Gal4, UAS-GFP/UAS-not^IR_.

**Video 3.** **Time-lapse movie of normal BC migration (10 frames/s) showing onset of migration, including the ability of cluster to acquire forward actin protrusions.** Protrusions at front, side, and back are 55.9%, 11.8%, 32.3%, respectively. MARCM clones are labeled with nuclear GFP; F-actin is labeled with LifeAct-GFP. Egg chamber genotype: _hsFLP, tub-Gal4, UAS-GFP/+; slbo-LifeAct-GFP/+; +, FRT80B/tub-Gal80, FRT80B._

**Video 4.** **Time-lapse movie of abnormal BC migration (10 frames/s) showing early tumbling of the cluster and multidirectional actin protrusions in _not^1_ cells labeled with nuclear GFP using MARCM.** In this example of a cluster that fails to delaminate, the percent of protrusions at front, side, and back are 13.1, 79.5, 7.4, respectively. The tumbling index is 100%. F-actin is labeled with LifeAct-GFP. Egg chamber genotype: _hsFLP, tub-Gal4, UAS-GFP/+; slbo-LifeAct-GFP/+; not^1, FRT80B/tub-Gal80, FRT80B._

**Video 5.** **Time-lapse movie of abnormal BC migration (10 frames/s) showing early tumbling of the cluster, loss of direction, and multidirectional actin protrusions in _not^1_ cells labeled with nuclear GFP using MARCM.** In this example of a cluster that delaminates but fails to fully migrate, the percent of protrusions at front, side, and back are 9.5, 62.0, and 28.5, respectively, similar to the mean distributions reported in Fig. 4 for clusters that fail to effectively delaminate. The tumbling index is 100%. F-actin is labeled with LifeAct-GFP. Egg chamber genotype: _hsFLP, tub-Gal4, UAS-GFP/+; slbo-LifeAct-GFP/+; not^1, FRT80B/tub-Gal80, FRT80B._

**Table S1, provided online as a separate Word file, summarizes data showing the association of Not and Ada2b with the promoters of Hippo pathway genes. Table S2, provided online as a separate Excel file, summarizes RNAi lines used to screen for _Drosophila_ DUBs involved in BC migration.**

