## [Peer Review File · The Journal of Cell Biology]

Drosophila USP22/non-stop polarises the actin cytoskeleton during collective border cell migration

Hammed Badmos, Neville Cobbe, Amy Campbell, Richard Jackson, and Daimark Bennett

Corresponding Author(s): Daimark Bennett, University of Liverpool

Review Timeline:

Submission Date:	2020-07-02
Editorial Decision:	2020-09-15
Revision Received:	2021-02-06
Editorial Decision:	2021-03-24
Revision Received:	2021-04-07
Editorial Decision:	2021-04-14
Revision Received:	2021-04-21

Monitoring Editor: Anna Huttenlocher

Scientific Editor: Dan Simon

Transaction Report:

DOI: <https://doi.org/10.1083/jcb.202007005>

September 15, 2020

Re: JCB manuscript #202007005

Dr. Daimark Bennett
University of Liverpool
Molecular Physiology and Cell Signalling
Institute of Systems Molecular and Integrative Biology
Crown Street
Liverpool L69 7ZB
United Kingdom

Dear Dr. Bennett,

Thank you for submitting your manuscript entitled "Drosophila USP22/non-stop regulates the Hippo pathway to polarise the actin cytoskeleton during collective border cell migration". Please accept our apologies for the delay in the processing of your manuscript. The manuscript was assessed by expert reviewers, whose comments are appended to this letter. Overall, the reviewers are enthusiastic about the study and we invite you to submit a revision. However, they raised several concerns which will need to be addressed before this paper would be ready for publication in JCB.

You will see that reviewers request quantifications for several experiments, verification of the partial rescue effects with another Not RNAi line, more data for the conclusion that polar cells arise from excessive proliferation, testing another component of the HAT module, as well as additional details in method descriptions and changes to text & figures. We feel that all of these requests are reasonable and should be addressed.

Reviewers also have concerns regarding the conclusion that non-stop regulates the Hippo pathway and suggest several experiments to provide more definitive evidence for this. We agree that these would be interesting, but as the core finding of this paper is that non-stop plays a role in border cell migration, we feel these are beyond the scope of this paper and not required for resubmission. However, if you decide not to provide additional data supporting the proposed regulation of Hippo pathway by non-stop then please tone down conclusions regarding this and change the title as requested by reviewers.

GENERAL GUIDELINES:

Text limits: Character count for an Article is < 40,000, not including spaces. Count includes title page, abstract, introduction, results, discussion, acknowledgments, and figure legends. Count does not include materials and methods, references, tables, or supplemental legends.

Figures: Articles may have up to 10 main text figures. Figures must be prepared according to the policies outlined in our Instructions to Authors, under Data Presentation, <http://jcb.rupress.org/site/misc/ifora.xhtml>. All figures in accepted manuscripts will be screened prior to publication.

IMPORTANT: It is JCB policy that if requested, original data images must be made available. Failure to provide original images upon request will result in unavoidable delays in publication. Please ensure that you have access to all original microscopy and blot data images before submitting your revision.

Supplemental information: There are strict limits on the allowable amount of supplemental data. Articles may have up to 5 supplemental figures. Up to 10 supplemental videos or flash animations are allowed. A summary of all supplemental material should appear at the end of the Materials and methods section.

As you may know, the typical timeframe for revisions is three to four months. However, we at JCB realize that the implementation of social distancing and shelter in place measures that limit spread of COVID-19 also pose challenges to scientific researchers. Lab closures especially are preventing scientists from conducting experiments to further their research. Therefore, JCB has waived the revision time limit. We recommend that you reach out to the editors once your lab has reopened to decide on an appropriate time frame for resubmission. Please note that papers are generally considered through only one revision cycle, so any revised manuscript will likely be either accepted or rejected.

Thank you for this interesting contribution to Journal of Cell Biology. You can contact us at the journal office with any questions, cellbio@rockefeller.edu or call (212) 327-8588.

Sincerely,

Anna Huttenlocher, M.D.
Monitoring Editor Journal
of Cell Biology

Dan Simon, Ph.D.
Scientific Editor
Journal of Cell Biology

Reviewer #1 (Comments to the Authors (Required)):

This manuscript characterises the role of USP22 in collective cell migration, where they show that it is required for Drosophila border cell migration and for expression of the Hippo pathway components Ex and Mer. Consequently, there is a defect in Hippo signalling and failure to polarise Crb to the inside of the cluster while F-actin fails to localise properly to the periphery of the border

cell cluster. The authors identify direct binding of USP22 to the promoters of *ex* and *mer*. Overall, the experiments are well performed and the manuscript should be published in JCB after attention to the following minor points:

1. Could it be that Yki recruits USP22 to the promoters of *ex* and *mer*? As the authors point out on page 10, in many contexts, Yki activates expression of *ex* and *mer*. The authors cite Lin et al 2014, but I wonder if they ought to repeat the experiment with (*actin>flipout>Gal4*, *slbo.Gal4*, *GR1.Gal4*, or *tj.Gal4*-driven) Yki-RNAi to be sure that Yki is not required for expression of *ex* and *mer*, or for border cell migration itself. The experiments shown in Lin et al are only partial clones of *yki* (half a cluster mutant) and *upd.Gal4* experiments, which do not express in the entire cluster.
2. Which E3 ubiquitin ligase does USP22 antagonise in border cells? Can the authors speculate? Would it be feasible to screen for rescue the USP22-RNAi phenotype by co-depletion of the cognate E3?
3. What is the phenotype of USP22-RNAi or *not1* mutation in the *Drosophila* wing or eye?

Reviewer #2 (Comments to the Authors (Required)):

Summary

This paper found a novel role for non-stop (*not*), which encodes a component of the Spt-Ada-Gcn5 Acetyltransferase (SAGA) chromatin-modifying complex, in border cell migration. The authors find that RNAi targeting of non-stop in border cells leads to severe migration defects and the mosaic clones mutant for non-stop cause cluster splitting, impaired migration, and increased polar and border cell numbers. Then, they explore mechanisms by which non-stop may impact border cell migration. They find that *not* is required to exclude F-actin from the interior of border cell cluster and to properly localize the polarity determinant, Crb. They find that mutation of non-stop does not reduce Scar and thus may not regulate the WAVE complex as had been reported in other contexts. Rather, non-stop regulates the expression of Expanded and Merlin in border cells. Their data and analysis of published ChIP seq data suggest that non-stop binds to expanded and merlin promoters independently of the SAGA complex.

This manuscript presents interesting preliminary findings identifying a novel role for non-stop in border cell development, which if substantiated could be of general interest. However, some of the conclusions are not fully supported by the data presented due to the low sample sizes, incomplete statistical analyses and the lack of quantification of the data presented in Figures 5-7. Further, the claim in the title that non-stop acts through the Hippo pathway requires substantiation. The authors would need to test the roles of core Hippo pathway components (e.g. Hippo, Warts, Yki) and determine epistasis between non-stop and the Hippo pathway components. As it stands the authors can conclude that non-stop regulates Expanded expression and that this accounts for some but not all of the effects in border cells. While Expanded can feed into the Hippo pathway, it is not certain that all effects of Expanded are mediated through the Hippo pathway, which is why it is important to test that directly. The authors also need to clarify which phenotypes are autonomous to which cell types (polar cells vs outer border cells).

Critique

The strengths of this study include the identification of a deubiquitinating enzyme required for border cell development, a convincing correlation between increased numbers of polar cells and a corresponding increase in border cell numbers, convincing changes in Expanded and Merlin

expression and in the localization of F-actin and polarity determinants in non-stop mutants (although quantification is needed for the results presented in Fig. 5-7).

The following weaknesses should be addressed:

1. Overall the sample sizes are low, statistical analyses are missing, and the variation for each genotype is not clearly displayed in figures. The authors should use the data that they have gathered to perform a power calculation and determine how many additional samples need to be analyzed in order for the differences between control and experimental conditions to reach statistical significance. For comparisons to be valid, the numbers of egg chambers analyzed for each genotype need to be similar (Fig 1H numbers vary from 37 to 101 and in 1J 47 and 138). They should clarify how many different times the crosses were performed (independent experiments), and how many flies were dissected per experiment, in addition to how many total egg chambers were analyzed. At least three independent experiments should be carried out for each genotype. It is critical to measure the variation within and between experiments in order to identify significant differences between different genotypes.
2. Could the authors provide a bit more detail about the screen they carried out? What lines were screened? What Gal4 lines used? How many caused a border cell phenotype?
3. It is very surprising that there is such a strong defect in the *slboGal4*, Not RNAi condition because *slboGal4* turns on immediately prior to migration, so does not really provide much time for RNAi to take effect and protein to turnover. This is a comment rather than a criticism, but it is a little hard to understand. Do the cells die?
4. The authors conclude based on partial rescue that 1) Not is required in outer border cells for migration and 2) the RNAi line may have an off-target effect, and this seems to be the reason that most of the analyses are shown using MARCM clones of a single mutant allele. Are there other RNAi lines against this gene that could confirm the requirement for Not in outer border cells? Did the authors control for the number of UAS transgenes in the experiment? This is important to verify that the partial rescue is not simply a result of titrating the Gal4, thus reducing the expression of the dsRNA. Do the authors have any idea what the identity of the "off-target" might be? Are there predictions available?
5. In Fig. 1C The appropriate control for Not overexpression is *slboGal4* rather than *c306*.
6. Is the split cluster phenotype in 1I due to loss in border cells or polar cells? Is it only observed in clusters with extra polar and/or border cells? Can it be rescued? If it is a polar cell autonomous phenotype, can it be replicated with *updGal4* and the not RNAi? What is the phenotype when only polar cells are mutant?
7. In figures 1F and 1G and in figures 2 and 3, different stages are shown for different genotypes. The authors should be careful when quantifying migration to ensure that they are analyzing the same stages for all genotypes.
8. When the authors compare *slbo-lacZ* levels between *not1* homozygous and heterozygous cells, in the clone shown in figure 2B, it looks like only the polar cells are GFP-. It is not appropriate to compare the level of *slbo-lacZ* in outer border cells to polar cells. It is important to analyze clones that contain both GFP+ and GFP- outer border cells.
9. It is unclear in Fig 2H which/how many cells are border cells.

10. Which not1 phenotypes are cell autonomous? Is the extra polar cell phenotype autonomous to the extra polar cell? To the polar cells as a group? More critically, which of the effects on border cells are due to an autonomous requirement for Not in the border cells and which are secondary consequences of the effects on polar cells? The increased border cell number is very likely due to increased polar cell number, and the authors state that larger clusters exhibit more severe migration defects, so the question becomes how severe are the migration defects due to loss of not in outer, migratory border cells alone? In most examples that are shown, many cells are homozygous mutant and polar cells are not labeled, so it is not possible to determine autonomy.

11. The authors conclude that excess polar cells arise from excessive proliferation but they do not provide evidence for this (phospho-histone H3 or cyclin B staining for example). Normally >2 polar cells develop and are eliminated by apoptosis. So it's possible that not1 mutant polar cells fail to undergo apoptosis. Or it could be that Not mutations cause extra precursors to adopt a polar cell fate. These possibilities cannot be distinguished based on the data provided.

12. In Figure 3B, please indicate where the line is that was used to quantify the signal. In Fig. 3C, do the terms "front, middle, and back" to the leading edge, middle, and trailing edge of the cluster? Do the data actually show a defect in front/back polarity? Or are front and back both meant to represent "outside" and the result meant to show a defect in inside/outside polarity? The effect on Crb suggests a defect in apical/basal polarity. Are all three polarities perturbed? In figure 3C, the graph does not show the variation between samples/error bars.

13. More quantification of Scar protein levels in mutant vs wild type cells and of F-actin localization are required in Figures 5 and 7 to draw conclusions. The conclusions that Scar is not reduced in not mutant cells, and that the scar and not mutant phenotypes are different, seem clear but it looks as though Scar levels might be higher in not mutant cells (though a mosaic cluster or looking at clones in the epithelium would provide a better comparison of wild type and mutant cells than comparing whole clusters in different egg chambers, which is what is shown in Fig. 5). The conclusion that non-stop acts independently of Scar is not fully supported since it could be that the increased levels of Scar contribute to the migration phenotype. Can Scar knockdown suppress any part of the not1 phenotype? Additionally, non-stop may affect border cell migration by altering the levels of hundreds of proteins either due to changes in transcription (like Expanded and Merlin) or by affecting protein stability, or due to indirect effects.

14. Quantification of data shown in Figure 6 (except for Crb which is quantified and shown in Fig 8) would strengthen the analysis.

15. It is not really clear if nonstop regulates the Hippo pathway (which is stated in the title) - because the authors did not look at core pathway components such as hippo, warts, and yorki, in not1 mutants. Inputs to the Hippo pathway and outputs from it are numerous, redundant, and complex. The extra border cell phenotype in not1 mutants is actually in the opposite of the effect of loss of Wrts from polar cells. According to Lin et al., 2014, Hippo normally functions in polar cells to repress Yki, which represses Upd and thus Jak/Stat. So reduced Hippo signaling or hyperactivity of Yki in polar cells causes reduced Upd expression and reduced numbers of border cells whereas the authors report that loss of not causes increased border cell numbers. It is also true that reducing Hippo signaling causes an increase in the numbers of polar cells (like not1), but in this case this does not lead to an increase in the number of border cells. Hpo and Wrts are also required earlier in development to inhibit Notch and promote polar cell fate. These differences between Hpo/Wrts and not1 call into question whether Not is affecting the Hippo pathway. Unless the authors mean that

Not affects the Hippo pathway specifically within the outer, migratory cells but this was not explicitly tested.

16. The rescue experiments in figure 8 show that overexpression of Ex partially rescues some of the defects in the nonstop mutants which is interesting. However, these results are not convincing without a higher sample size and statistical analysis of border cell migration defects in Figure 8M.

17. Again, the graphs in Figure 8 K and L do not show the variability between samples in each condition but are just showing means

18. Can the authors say if the migration defects are due to cluster size versus F-actin distribution? Did they look to see in 8 if ex or cpb overexpression rescues cluster size defects as well?

19. The rescue is only partial in Figure 8. Could the authors try to perturb the binding sites for non-stop in ex and merlin to more directly test if it binding to these sites to promote border cell migration? Or confirm in their own hands the ChIP seq results?

Minor comments

20. " Later, F-actin accumulates around the cortex of the cluster, as cells alternate their position in the cluster as they move collectively (Bianco et al., 2007)." No evidence is shown of differential cortical actin accumulation in the first vs second phase of migration proposed by the authors. If this is the case, then the analysis in Fig 3 should be done according to the "phase" of migration.

21. It would help the uninitiated reader if the authors provide some diagrams illustrating the protein complexes and pathways analyzed in the manuscript.

Reviewer #3 (Comments to the Authors (Required)):

The manuscript by Badmos and colleagues describes the identification of the Usp22 homolog Non-stop (Not) as a new regulator of border cell collective migration in the Drosophila ovary. Not is a deubiquitinating enzyme (DUB) that functions in the histone H2B DUB module of the SAGA transcriptional coactivator complex. The authors first show that not is required for border cell migration, with most clusters failing to reach the oocyte by the correct stage. In some cases, the cluster splits apart. In addition, there is an increase in the number of polar cells with a corresponding increase in border cell number. While several markers of border cell fate were unchanged in not mutant border cells, F-actin localization was altered. Normally, F-actin is primarily localized to the cluster periphery (cortex), but in not mutant clusters more F-actin is now enriched at cell-cell junctions between border cells inside the cluster. Live imaging of not mutant clusters showed that border cells had less polarized protrusions and possibly early cluster "tumbling", suggesting less polarized motility. Not was recently shown to regulate the Arp2/3 and WAVE regulatory complex member Scar. However, Scar mutants have distinct phenotypes from not mutants, suggesting that these two proteins function independently in border cells. Not mutants have similar phenotypes to mutants in the Hippo pathway (Lucas et al., J Cell Biol 2013), including effects on F-actin and polar cell number. The authors propose that Not regulates the upstream Hippo pathway components Expanded and Merlin, and that this is independent of the SAGA HAT module member Ada2b. Interestingly, other polarity proteins such as Crumbs are mislocalized in not mutant border cells. Not may regulate expression of expanded and merlin, as Not protein binds to the transcription start site of both genes and impacts their expression in border cells. Finally, the authors overexpressed Expanded and a known downstream target of the Hippo pathway in border cells, Capping protein B

(CPB), and demonstrated partial rescue of border cell migration and polarization of the cluster.

Overall, this manuscript presents new findings on the role of Usp22/Not in regulating polarized collective cell migration, with new connections to the Hippo pathway through regulation of Expanded and Merlin. There are implications for the polarized migration of cell collectives. This manuscript is generally well-written and well-documented. Most of the data is quantified very clearly. However, some conclusions need additional support, with clarification of key findings.

1. In Figure 1I and J, can the authors clarify if split clusters occur typically when more cells are mosaic mutant for not within the cluster, or when fewer cells are mutant? In other words, are not mutant cells splitting from wild-type cells or from other not mutant cells.
2. The authors state (e.g. p. 2) that non-stop is "at the top of a regulatory network underlying collective migration." I am unsure that their data really show this.
3. In Figure 3, the data are convincing and the histograms in panel B are useful. Can the authors show a line in panel A images indicating where made line scans shown in panel B? Panel C should provide statistics similar to what is shown in Figure 8L.
4. Videos S1-S3, can the authors add time stamps and/or indicate the length of the video (and what is the frame speed)?
5. In the section starting on p. 4 (including Figure 4 results), the authors analyze their movies for protrusions and tumbling. The authors define the two phases of migration as the "initial polarized phase, and a second phase that utilizes collective migration.... cells alternate their position in the cluster as they move collectively." I disagree that the first phase is not collective, as the cluster is polarized as an entire entity with a single leading protrusion. I would suggest that the authors can just simply define the two phases, both of which have distinct collective behaviors.
6. Similarly, it would be helpful if the authors could further clarify what they consider to be "tumbling" behavior (and add this to the Methods). Border cells are known to switch places within the cluster at different times during migration (Prasad and Montell, *Dev Cell* 2007), but rotation or tumbling I believe is more at the cluster level with the entire cell group rotating around an axis (Bianco et al., *Nature* 2007; Poukkula et al., *J Cell Biol* 2011). The videos they show do not really illustrate this very clearly, especially the wild-type video S2. I am wondering if the behavior they are seeing in not mutant clusters is cell-cell exchange rather than premature tumbling? Moreover, do they see differences in these cluster behaviors if the not mutant cells partially migrated versus didn't migrate at all (such as shown in video S3)?
7. For Figure 6 data, I am convinced by changes in the patterns and/or levels of ex-lacZ, Crumbs, and aPKC in not mutant clusters versus wild type. I agree that Ena looks mostly similar between not mutants and wild type. However, the changes in Merlin are not very obvious in the images shown and should be quantified. Likewise, any changes to the Armadillo localization are not that clear. All of the data in this figure should be quantified as much as possible, and N's reported as the authors do in other figures within this manuscript. It appears that the Crumbs mean intensity is reported in Figure 8K - something similar for the rest of these markers could be included to be more convincing.
8. The authors find that Not binds to the expanded and merlin promoters (ChIPSeq from Li et al., 2017; Figure 7A and 7B). Did the authors check for Not ChIPSeq to the start sites of other gene members of the Hippo pathway, including Crumbs?

9. The authors show that Ada2b, a SAGA-specific HAT module subunit is not required for border cell migration nor impacts ex-lacZ or F-actin. How definitive is this data, having only used one mutant allele? Is it possible to test another component of the HAT module to confirm this result? Also, in Figure 7C, is ex-lacZ higher in mutant cells?

10. In the last section of the results (p. 8), the authors conclude that "expanded is a critical transcriptional target of non-stop required for its function in border cells." I would argue that their data support expanded (and Cpb) as being targets/downstream, but either there are other parallel downstream targets or technical reasons that neither Expanded nor Cpb strongly rescue the not mutant phenotypes. There is only mild rescue of migration (Figure 8M; the authors should add statistics here). From their data, it seems that Expanded has more impact on rescuing Crumbs localization (Figure 8G and K), whereas Cpb seems to have a greater rescue on F-actin localization (Figure 8J and L).

11. Figure 8L, the authors could show statistics comparing the phenotypes of the rescues to the not mutant border cells, not just to wild type.

12. It is intriguing how not mutants strongly impact Crumbs localization and F-actin in border cell clusters. In the discussion (p. 10 "non-stop regulates the distribution of polarity determinants"), they mention possible relationship to Moesin. This may be beyond the scope of this manuscript, but did the authors look at Moesin localization in not mutants (e.g. Moesin localization as shown in Ramel et al. Nat Cell Biol 2013)? The authors discuss a recent paper that showed that Moesin stabilizes Crumbs (Aguilar-Aragon et al., 2020). Likewise, Ramel et al. (2013) showed that Moesin regulates F-actin organization and polarity of the border cell cluster. Could an effect on Moesin explain both the Crumbs and F-actin effects by Non-stop?

Minor comment:

1. Missing call out to figure on p. 5 "Polarisation of the polarity determinant Crb...." (Figure 5E).

RESPONSE

Please find our point-by-point responses to the reviewers' comments below.

September 15, 2020

Re: JCB manuscript #202007005

Dr. Daimark Bennett
University of Liverpool
Molecular Physiology and Cell Signalling
Institute of Systems Molecular and Integrative Biology
Crown Street
Liverpool L69 7ZB
United Kingdom

Dear Dr. Bennett,

Thank you for submitting your manuscript entitled "Drosophila USP22/non-stop regulates the Hippo pathway to polarise the actin cytoskeleton during collective border cell migration". Please accept our apologies for the delay in the processing of your manuscript. The manuscript was assessed by expert reviewers, whose comments are appended to this letter. Overall, the reviewers are enthusiastic about the study and we invite you to submit a revision. However, they raised several concerns which will need to be addressed before this paper would be ready for publication in JCB.

You will see that reviewers request quantifications for several experiments, verification of the partial rescue effects with another Not RNAi line, more data for the conclusion that polar cells arise from excessive proliferation, testing another component of the HAT module, as well as additional details in method descriptions and changes to text & figures. We feel that all of these requests are reasonable and should be addressed.

Reviewers also have concerns regarding the conclusion that non-stop regulates the Hippo pathway and suggest several experiments to provide more definitive evidence for this. We agree that these would be interesting, but as the core finding of this paper is that non-stop plays a role in border cell migration, we feel these are beyond the scope of this paper and not required for resubmission. However, if you decide not to provide additional data supporting the proposed regulation of Hippo pathway by non-stop then please tone down conclusions regarding this and change the title as requested by reviewers.

GENERAL GUIDELINES:

Text limits: Character count for an Article is < 40,000, not including spaces. Count includes title page, abstract, introduction, results, discussion, acknowledgments, and figure legends. Count does not include materials and methods, references, tables, or supplemental legends.

Figures: Articles may have up to 10 main text figures. Figures must be prepared according to the policies outlined in our Instructions to Authors, under Data Presentation, <http://jcb.rupress.org/site/misc/ifora.xhtml>. All figures in accepted manuscripts will be screened prior to publication.

Supplemental information: There are strict limits on the allowable amount of supplemental data. Articles may have up to 5 supplemental figures. Up to 10 supplemental videos or flash animations are allowed. A summary of all supplemental material should appear at the end of the Materials and methods section.

As you may know, the typical timeframe for revisions is three to four months. However, we at JCB realize that the implementation of social distancing and shelter in place measures that limit spread of COVID-19 also pose challenges to scientific researchers. Lab closures especially are preventing scientists from conducting experiments to further their research. Therefore, JCB has waived the revision time limit. We recommend that you reach out to the editors once your lab has reopened to decide on an appropriate time frame for resubmission. Please note that papers are generally considered through only one revision cycle, so any revised manuscript will likely be either accepted or rejected.

Thank you for this interesting contribution to Journal of Cell Biology. You can contact us at the journal office with any questions, cellbio@rockefeller.edu or call (212) 327-8588.

Sincerely,

Anna Huttenlocher, M.D.
Monitoring Editor
Journal of Cell Biology

Dan Simon, Ph.D.
Scientific Editor
Journal of Cell Biology

Reviewer #1 (Comments to the Authors (Required)):

This manuscript characterises the role of USP22 in collective cell migration, where they show that it is required for *Drosophila* border cell migration and for expression of the Hippo pathway components *Ex* and *Mer*. Consequently, there is a defect in Hippo signalling and failure to polarise *Crb* to the inside of the cluster while F-actin fails to localise properly to the periphery of the border cell cluster. The authors identify direct binding of USP22 to the promoters of *ex* and *mer*. Overall, the experiments are well performed and the manuscript should be published in JCB after attention to the following minor points:

1. Could it be that *Yki* recruits USP22 to the promoters of *ex* and *mer*? As the authors point out on page 10, in many contexts, *Yki* activates expression of *ex* and *mer*. The authors cite Lin et al 2014, but I wonder if they ought to repeat the experiment with (actin>flipout>Gal4, *slbo*.Gal4, GR1.Gal4, or *tj*.Gal4-driven) *Yki*-RNAi to be sure that *Yki* is not required for expression of *ex* and *mer*, or for border cell migration itself. The experiments shown in Lin et al are only partial clones of *yki* (half a cluster mutant) and *upd*.Gal4 experiments, which do not express in the entire cluster.

Lack of migration defects in *yki* loss of function argues against there being an essential role for *Yki* in recruiting Not/USP22 to the promoters of *ex* and *mer*. To increase our confidence in this conclusion, we have performed *slbo*-GAL4, *UAS-yki* RNAi experiments as suggested. We find that *yki* RNAi (at 30°C, using a widely-used inverted repeat line) in the outer border cells fails to abrogate migration (Fig.S3), in agreement with the conclusion of Lin et al that *yki* is dispensable for border cell migration.

2. Which E3 ubiquitin ligase does USP22 antagonise in border cells? Can the authors speculate? Would it be feasible to screen for rescue the USP22-RNAi phenotype by co-depletion of the cognate E3?

A genetic screen for E3 ubiquitin ligases that suppress *not*/USP22 loss of function could be an effective way to identify antagonistic enzymes. Enzymes targeting histone H2B for ubiquitination, such as Bre1, would be the most likely candidates since ubiquitinated histone is the canonical target for Not/USP22. However, we do not consider this to be a satisfactory approach to assess the unique and overlapping contribution of the E3 ligases because the *not* RNAi does not present a clean phenotype (due to off-target effects). It may be possible to engineer a suitable tester strain for a screen using CRISPR-Cas9 (see Fig.S1), but this is complicated by additional genetic elements needed for this system to work. Due to difficulties importing *Drosophila* strains into the UK because of Covid19 and Brexit, we have been unable to develop this line of investigation further at this time.

3. What is the phenotype of USP22-RNAi or not1 mutation in the Drosophila wing or eye?

Preliminary experiments in which we have generated negatively marked *not¹* mutant clones in the wing or eye suggest that *non-stop* has little or no effect on clonal growth. This is in keeping with a report that knockdown of *non-stop* in the developing eye (using VDRC-45776, which is the line we have employed in our study) does not cause eye disc hypoplasia and does not modulate Ras-induced hyperplasia (Fernández-Espartero C *et al.* 2018, Development 145: dev162156). In response to Reviewer 2 we assessed the cell-autonomous requirement for *non-stop* in polar cells. Interestingly, in this context, inhibition of Yki by Hippo signalling is required to maintain border cell numbers via upd signalling (Lin *et al.* 2014), whereas *non-stop* is not (Fig.S1). This suggests that *non-stop* may be specifically required in contexts, including border cells, where the hippo complex signals to directly modulate the actin cytoskeleton via Ena, rather than in cells where canonical signalling functions to prevent Yki activation. We have clarified this issue in the discussion.

Reviewer #2 (Comments to the Authors (Required)):

Summary

This paper found a novel role for non-stop (not), which encodes a component of the Spt-Ada-Gcn5 Acetyltransferase (SAGA) chromatin-modifying complex, in border cell migration. The authors find that RNAi targeting of non-stop in border cells leads to severe migration defects and the mosaic clones mutant for non-stop cause cluster splitting, impaired migration, and increased polar and border cell numbers. Then, they explore mechanisms by which non-stop may impact border cell migration. They find that not is required to exclude F-actin from the interior of border cell cluster and to properly localize the polarity determinant, Crb. They find that mutation of non-stop

does not reduce Scar and thus may not regulate the WAVE complex as had been reported in other contexts. Rather, non-stop regulates the expression of Expanded and Merlin in border cells. Their data and analysis of published ChIP seq data suggest that non-stop binds to expanded and merlin promoters independently of the SAGA complex.

This manuscript presents interesting preliminary findings identifying a novel role for non-stop in border cell development, which if substantiated could be of general interest. However, some of the conclusions are not fully supported by the data presented due to the low sample sizes, incomplete statistical analyses and the lack of quantification of the data presented in Figures 5-7. Further, the claim in the title that non-stop acts through the Hippo pathway requires substantiation. The authors would need to test the roles of core Hippo pathway components (e.g. Hippo, Warts, Yki) and determine epistasis between non-stop and the Hippo pathway components. As it stands the authors can conclude that non-stop regulates Expanded expression and that this accounts for some but not all of the effects in border cells. While Expanded can feed into the Hippo pathway, it is not certain that all effects of Expanded are mediated through the Hippo pathway, which is why it is important to test that directly. The authors also need to clarify which phenotypes are autonomous to which cell types (polar cells vs outer border cells).

Critique

The strengths of this study include the identification of a deubiquitinating enzyme required for border cell development, a convincing correlation between increased numbers of polar cells and a corresponding increase in border cell numbers, convincing changes in Expanded and Merlin expression and in the localization of F-actin and polarity determinants in non-stop mutants (although quantification is needed for the results presented in Fig. 5-7).

The following weaknesses should be addressed:

1. Overall the sample sizes are low, statistical analyses are missing, and the variation for each genotype is not clearly displayed in figures.

We have taken this opportunity to improve our graphs throughout, using SuperPlots (Lord et al JCB 2020 219(6):e202001064) where appropriate, to communicate the cell-level variability and experimental reproducibility. Statistical analyses are now provided throughout. In each case, we have taken the approach recommended by Lord et al, which is to compare the mean measurements from samples in each biological repeat (n of at least 3) rather than pooling all measurements together.

The authors should use the data that they have gathered to perform a power calculation and determine how many additional samples need to be analyzed in order for the differences between control and experimental conditions to reach statistical significance.

We have enlisted the help of Dr Richard Jackson (Acting Head of Statistics, Cancer Division, University of Liverpool), to assist us with power calculations. Given the large number of comparisons made, we have taken the outcome with the largest standard deviation, which in this case is the data relating to figure 1H. We take a biologically meaningful difference to be change in a given outcome represented by a log(2)-fold change of 1. Measuring outcomes on the log scale gives an observed standard

deviation of 0.78. Here using an alpha level of 0.01 to account for multiple testing and ensuring a family wise error rate preserved below 0.05 and including 90% power, only 13 observations in each group are required with the small number obtained due to the relatively large difference which determines biological significance. Correspondingly, for intensity measurements, smaller numbers of observations are required where the biologically meaningful difference is determined by a larger log(2)-fold change.

For comparisons to be valid, the numbers of egg chambers analyzed for each genotype need to be similar (Fig 1H numbers vary from 37 to 101 and in 1J 47 and 138).

We would not agree that disparity in effect numbers will affect the validity of any comparisons. Whilst we agree that any imbalance in numbers will negatively impact the power of any comparison and will lead to a loss in precision when making comparisons, we do not recognise that this leads to a loss of 'validity'. The comparisons themselves are still protected from undue sources of bias, which are protected against by the conduct of a prospectively designed experimental study and the differences reported can be interpreted in that light. Indeed, in the field of clinical medicine, larger trials are often designed with unequal allocation without any concern to the validity of the results.

They should clarify how many different times the crosses were performed (independent experiments), and how many flies were dissected per experiment, in addition to how many total egg chambers were analyzed. At least three independent experiments should be carried out for each genotype.

Crosses (independent experiments) were done at least 3 times, with the number of flies dissected per experiment varying according to the nature of the experiment and availability of material – e.g. not all trials of clonal experiments yielded suitable mosaics. We have provided specific details in the legends to each figure and included additional information in the Methods.

It is critical to measure the variation within and between experiments in order to identify significant differences between different genotypes.

See our response to this point above.

2. Could the authors provide a bit more detail about the screen they carried out? What lines were screened? What Gal4 lines used? How many caused a border cell phenotype?

We screened 133 different transgenic RNAi lines, representing the 45 *Drosophila* DUBs, initially using *slbo-GAL4*, *UAS-GFP* in the presence of *UAS-Dcr2* (to nominally enhance RNAi knockdown and increase screen sensitivity). We then repeated crosses using *slbo-GAL4* in the absence of *UAS-Dcr2* (for increased specificity in case Dcr2 increased off-target knockdown effects) as well as *c306-GAL4*; *UAS-GFP*, *UAS-DsRed^{NLS}*. Only *non-stop* showed a comparable number of defects to a known regulator of invasive migration that we used as our positive control (*msn*; Cobrerros-Reguera L et al. 2010 EMBO Reports). We have included a table of RNAi lines that were screened (Table S2).

3. It is very surprising that there is such a strong defect in the *slboGal4*, Not RNAi condition because *slboGal4* turns on immediately prior to migration, so does not really

provide much time for RNAi to take effect and protein to turnover. This is a comment rather than a criticism, but it is a little hard to understand. Do the cells die?

Whilst it is perhaps surprising to observe such a strong effect with *slbo-GAL4*, it is not without precedent; indeed, we benchmarked hits in our original screen against *msn* knockdown, which also strongly abrogated migration. It is true that knockdown of many other genes involved in migration does not have such a strong effect; this presumably is down to the degree of knockdown, turnover rates and the point at which protein levels become rate limiting. We don't see any evidence of cell death in *slbo>not^{IR}* border cells. For the benefit of both the reviewer and reader, we have included a time-lapse video showing defective border cell migration in *slbo>not^{IR}* flies (Video S2).

4. The authors conclude based on partial rescue that 1) Not is required in outer border cells for migration and 2) the RNAi line may have an off-target effect, and this seems to be the reason that most of the analyses are shown using MARCM clones of a single mutant allele. Are there other RNAi lines against this gene that could confirm the requirement for Not in outer border cells?

We did identify a second RNAi line (VDRC 45775) in our primary screen that appeared to weakly reduce migration with *slbo-GAL4*, but this did not pass secondary screening to validate the effect. The TRiP line JF03152 similarly has a weak effect. To further confirm the requirement for *non-stop* in the border cell cluster, we have turned to a tissue-specific CRISPR/Cas9 approach (Port and Bullock, Nat Methods 2016; Port *et al* PNAS 2014). For this, we used a transgene that constitutively expresses two single guide RNAs targeting *non-stop* immediately downstream of the translation initiation site, in combination with *UAS-Cas9*. Expression of *not^{sgRNA}* with *Cas-9* in all cells in the cluster (with *c306-GAL4*) strongly abrogated migration, expression in outer border cells from a later stage of development (with *slbo-GAL4*) had a less pronounced effect, whereas expression solely in polar cells (with *upd-GAL4*) had no effect (Fig.S1). We have also confirmed the specific requirement in outer border cells by a closer analysis of a subset of *not¹* clones stained for the polar cell marker *Fas3*. Importantly, there was no significant difference between the degree of migration of border cell clusters that were either entirely *not¹* mutant (including polar cells) or of clusters that were >80% *not¹* mutant (and contained wild type polar cells), Fig.1H. See also Point 10 for a discussion of (non-) cell autonomous effects.

Did the authors control for the number of UAS transgenes in the experiment? This is important to verify that the partial rescue is not simply a result of titrating the Gal4, thus reducing the expression of the dsRNA.

With regard to controls for UAS transgenes, we have tested this previously, but did not include the data in the original submission. We find that unlike *UAS-not^{*}* an additional *UAS-GFP* line does not substantially rescue *UAS-not^{IR}* (see revised Fig.1E).

Do the authors have any idea what the identity of the "off-target" might be? Are there predictions available?

Since we are at the limit of supplementary online items that we can provide, please find a readout of on and potential off-targets for the *not* RNAi line, which was generated using the online Up-Torr search engine (<https://www.flyrnai.org/up-torr/>), at the end of this response. This table also includes some information about the predicted function of potential off-targets that we curated from Flybase.

5. In Fig. 1C The appropriate control for Not overexpression is *slboGal4* rather than *c306*.

This is a good point, we have replaced the *c306>not^{tr}* with a *slbo>not^{tr}* control in Fig1C and E.

6. Is the split cluster phenotype in 1I due to loss in border cells or polar cells?

Two lines of investigation suggest the split cluster phenotype is due to loss in border cells: firstly, clusters that contain mutant outer border cells but wild type polar cells display the splitting phenotype (Fig.1I, 2C); secondly, targeting *non-stop* in polar cells with CRISPR-cas9 has no such effect (Fig.S1, see also responses below).

What is the phenotype when only polar cells are mutant?

We have not been able to generate clones that solely affect the polar cells. However, targeting *non-stop* in polar cells with CRISPR-Cas9 has little or no effect. This issue is also raised in point 10, where we provide a more comprehensive response.

Is it only observed in clusters with extra polar and/or border cells?

Splitting is not limited to larger clones containing extra cells – the relationship between size and splitting is illustrated in Fig.2F and discussed in the revised text. A correlation matrix with calculated R values is given below. As this shows, there is not a strong correlation between number of total nuclei in the cluster and frequency of splitting, or between proportion of *not¹* cells and frequency of splitting. There is a moderate correlation (0.63) between the total number of nuclei and the proportion of *not¹* cells.

	Proportion of not¹ cells	Frequency of splitting
Frequency of splitting	0.00930513	
Total nuclei	0.63299014	-0.01289982

Can it be rescued?

Splitting can be rescued by *not* overexpression; we have included these data in Fig.1J.

If it is a polar cell autonomous phenotype, can it be replicated with updGal4 and the not RNAi?

It is not a polar cell autonomous phenotype. See comment above and response to Point 10 below.

7. In figures 1F and 1G and in figures 2 and 3, different stages are shown for different genotypes. The authors should be careful when quantifying migration to ensure that they are analyzing the same stages for all genotypes.

The different stages in Figure 1 are shown for illustrative purposes, but quantitation was only ever done on egg chambers of the same stage. Some panels have been replaced in the revised figure to avoid confusion.

8. When the authors compare *slbo-lacZ* levels between *not1* homozygous and heterozygous cells, in the clone shown in figure 2B, it looks like only the polar cells are GFP-. It is not appropriate to compare the level of *slbo-lacZ* in outer border cells to polar cells. It is important to analyze clones that contain both GFP+ and GFP- outer border cells.

We agree, the image we originally selected appears misleading – quantification of *slbo-lacZ* (panel C) was only done for GFP+ and GFP- outer border clones in the same cluster. We have replaced the image in panel B with one that illustrates this more

clearly, and checked our quantification to confirm only outer border cells were being compared in the analysis reported in Fig.2.

9. It is unclear in Fig 2H which/how many cells are border cells.

We agree that this image was not presented in the best possible way. We have now included a panel showing segmentation of (*upd-lacZ* labelled) polar and (GFP-labelled) border cells (Fig.2D').

10. Which *not1* phenotypes are cell autonomous? Is the extra polar cell phenotype autonomous to the extra polar cell? To the polar cells as a group? More critically, which of the effects on border cells are due to an autonomous requirement for Not in the border cells and which are secondary consequences of the effects on polar cells? The increased border cell number is very likely due to increased polar cell number, and the authors state that larger clusters exhibit more severe migration defects, so the question becomes how severe are the migration defects due to loss of *not* in outer, migratory border cells alone? In most examples that are shown, many cells are homozygous mutant and polar cells are not labeled, so it is not possible to determine autonomy.

The reviewer raises an important issue here regarding autonomous/non-autonomous effects. This issue has been partially addressed in our responses to the points above, but is expanded upon here and in our response to the Point 11, below. To address this issue, we took two complementary approaches.

- 1) We stained egg chambers harbouring *not1* mutant clones with either Fas3 or *upd-lacZ*, to label polar cells. Although we were unable to observe border cell clusters in which polar cells alone were mutant, we were able to generate clusters in which only outer border cells were mutant (e.g. Fig.1G). These demonstrated strong migration defects, consistent with a role for *non-stop* in outer border cells.
- 2) We targeted *non-stop* with CRISPR-Cas9 in outer border cells alone, polar cells alone or the whole cluster, as described in our response to point 4 above. These data point to a role for *non-stop* solely in outer border cells, and, together with analysis of *not1* clones (see Point 11 below), identify a non-cell autonomous role for *non-stop* in controlling polar cell number.

Extra polar cells that result from non-cell autonomous effects are correlated with the increase in border cell number (Fig2A); does this increase in cluster size impact on migration? To address this, we plotted the relationship between cluster size (total number of nuclei) and % *not1* cells and % migration (Fig.2E). Strong migration defects are clearly observed in *not1* mutant clusters with normal numbers of cells (6-8 cells), so migration defects are not dependent on the generation of larger clusters. Further exploratory regression analysis including the total number of nuclei as a response and % migration and % *not1* cells as explanatory covariates suggests there is, at best, a weak negative correlation between total number of nuclei and % migration. A correlation matrix with calculated R values is given below. This might suggest that there is some impairment to migration when clusters are larger, possibly due to topological constraints on migration as they pass through the nurse cell milieu, but this is not the main factor determining rate of migration.

	Total nuclei	% migration
% migration	-0.4318811	
% not ¹ clone	0.2769014	-0.436042

11. The authors conclude that excess polar cells arise from excessive proliferation but they do not provide evidence for this (phospho-histone H3 or cyclin B staining for example). Normally >2 polar cells develop and are eliminated by apoptosis. So it's possible that *not1* mutant polar cells fail to undergo apoptosis. Or it could be that *Not* mutations cause extra precursors to adopt a polar cell fate. These possibilities cannot be distinguished based on the data provided.

We're grateful that the reviewer raised this issue, which we did not give enough attention to in our original submission. As we have now established (see responses above), *non-stop* non-autonomously affects polar cell number, but the impact of this on migration (via effects on cluster size) are fairly minimal. Some preliminary analysis of stage 10 egg chambers shows that extra cells in the cluster are not associated with cyclin B staining, but there are a number of possibilities that could explain the extra polar cell phenotype we have uncovered. Although it will be interesting to learn the basis of this phenomenon, we believe a detailed analysis of the non-autonomous effect of *not*¹ clones on polar cell number is beyond the scope of our current study, and has no bearing on primary focus of this manuscript which is on *non-stop*'s role in collective cell migration.

12. In Figure 3B, please indicate where the line is that was used to quantify the signal. In Fig. 3C, do the terms "front, middle, and back" to the leading edge, middle, and trailing edge of the cluster? Do the data actually show a defect in front/back polarity? Or are front and back both meant to represent "outside" and the result meant to show a defect in inside/outside polarity? The effect on *Crb* suggests a defect in apical/basal polarity. Are all three polarities perturbed? In figure 3C, the graph does not show the variation between samples/error bars.

"Front, middle, and back" do indeed refer to the leading edge, middle, and trailing edge of the cluster; we have clarified this in the text. In our statistical comparisons we have compared the changes in the relative intensity in the middle of the cluster, and consequently our conclusions in the Discussion refer to outside/in polarity. The reviewer rightly identifies changes in *Crb* and *aPKC* as being indicative of effects on apical/basal polarity that we have commented on in the Discussion, and may account for the changes in orientation of protrusions that we have observed. We have replotted the data in Fig.3C to show variation in individual values.

13. More quantification of *Scar* protein levels in mutant vs wild type cells and of F-actin localization are required in Figures 5 and 7 to draw conclusions. The conclusions that *Scar* is not reduced in *not* mutant cells, and that the *scar* and *not* mutant phenotypes are different, seem clear but it looks as though *Scar* levels might be higher in *not* mutant cells (though a mosaic cluster or looking at clones in the epithelium would provide a better comparison of wild type and mutant cells than comparing whole clusters in different egg chambers, which is what is shown in Fig. 5). The conclusion that *non-stop* acts independently of *Scar* is not fully supported since it could be that the increased levels of *Scar* contribute to the migration phenotype. Can *Scar* knockdown suppress any part of the *not1* phenotype? Additionally, *non-stop* may affect border cell migration by altering the levels of

hundreds of proteins either due to changes in transcription (like Expanded and Merlin) or by affecting protein stability, or due to indirect effects.

Our primary objective here was to assess the possibility that *not* promotes border cell migration by protecting Scar from degradation, since it has been proposed that *not* can perform this role in a different context (Cloud *et al.*, 2019). Quantification of Scar levels in the epithelium confirm that there is not a reduction in Scar levels in *non-stop* mutant cells compared to their wild type siblings (Fig.5A' and B') – indeed, there is no significant change (Fig.5D). However, the reviewer is right to highlight that Scar may nevertheless contribute to aberrant actin accumulation at border cell-border cell junction. Indeed, it might be expected that along with actin polymerisation, promoted by Ena activation, Scar would be recruited to promote filament branching. We have explored the possibility that Scar is involved by carefully quantifying the distribution of Scar in the border cell cluster as well as performing complementation experiments with *scar* RNAi, as suggested. These experiments suggest there is a modest accumulation of Scar in border cell-border cell junctions that contributes somewhat to the effects on actin polarity (Fig.5F, K). However, we found that *scar* RNAi did not significantly suppress the effect of *not* loss of function on border cell migration (Fig.5G). We have also strengthened this section with analysis of the phenotypic effects of Scar overexpression, which show that altered distribution of Scar, rather than increased levels *per se*, contributes to its modest effect on actin polarity. See revised text and Fig.5. The referee is right in saying that other targets of *not* may affect migration. We alert readers to this issue in the conclusion and the legend to our schematic illustration, Fig.9.

14. Quantification of data shown in Figure 6 (except for Crb which is quantified and shown in Fig 8) would strengthen the analysis.

We have included quantification for these data in Figure 6.

15. It is not really clear if nonstop regulates the Hippo pathway (which is stated in the title) - because the authors did not look at core pathway components such as hippo, warts, and yorki, in *not1* mutants. Inputs to the Hippo pathway and outputs from it are numerous, redundant, and complex. The extra border cell phenotype in *not1* mutants is actually in the opposite of the effect of loss of Wrts from polar cells. According to Lin *et al.*, 2014, Hippo normally functions in polar cells to repress Yki, which represses Upd and thus Jak/Stat. So reduced Hippo signaling or hyperactivity of Yki in polar cells causes reduced Upd expression and reduced numbers of border cells whereas the authors report that loss of *not* causes increased border cell numbers. It is also true that reducing Hippo signaling causes an increase in the numbers of polar cells (like *not1*), but in this case this does not lead to an increase in the number of border cells. Hpo and Wrts are also required earlier in development to inhibit Notch and promote polar cell fate. These differences between Hpo/Wrts and *not1* call into question whether Not is affecting the Hippo pathway. Unless the authors mean that Not affects the Hippo pathway specifically within the outer, migratory cells but this was not explicitly tested.

[Also, as mentioned in the opening comments from the reviewer: "Further, the claim in the title that non-stop acts through the Hippo pathway requires substantiation."]

We would argue that *not* regulates non-canonical hippo complex activity in so much as *not* i) is required for the normal expression of *ex* and *mer*, which help tether hippo

to *crb*, and ii) phenocopies F-actin polarity defects in *warts* (Lucas et al JCB 2013) and, to a lesser extent, *ex* loss of function (Fig.8) in border cells. As discussed in our response to Reviewer 1, in polar cells, inhibition of Yki by Hippo signalling is required to maintain border cell numbers (Lin et al. 2014), whereas our analysis using CRISPR-Cas9 indicate that *non-stop* does not (Fig.S1). This suggests that *non-stop* may be specifically required in contexts, including outer border cells, where the hippo complex signals to directly influence the actin cytoskeleton, rather than in cells where canonical signalling functions to prevent Yki activation. We agree more could be done to address this issue if we were in possession of the appropriate reagents and had the capacity to conduct these experiments. Therefore, in accordance with the Editor's suggestion, we propose to address this point with a revision to the title, focussing on the effect of *not* on the actin cytoskeleton: "*Drosophila* USP22/*non-stop* polarises the actin cytoskeleton during collective border cell migration". We have also edited the text to clarify this issue.

16. The rescue experiments in figure 8 show that overexpression of *Ex* partially rescues some of the defects in the *nonstop* mutants which is interesting. However, these results are not convincing without a higher sample size and statistical analysis of border cell migration defects in Figure 8M.

We have included additional data here to satisfy power calculations, as performed by our statistician to give 90% power. Sample size and stats are provided in the text or shown on the figures. Please see also our responses to Point 1 regarding the approach we have taken to statistical analyses.

17. Again, the graphs in Figure 8 K and L do not show the variability between samples in each condition but are just showing means

We have now replotted the data with individual data points that show the variability between samples in each condition with the mean (\pm SE) of the means from each biological repeat marked in each case.

18. Can the authors say if the migration defects are due to cluster size versus F-actin distribution?

This issue has been addressed in a previous point (Point 10).

Did they look to see in 8 if *ex* or *cpb* overexpression rescues cluster size defects as well?

Overexpression of *expanded*, and to a lesser extent overexpression of *cpb*, partially suppresses the increased size and splitting of *non-stop* mutant clusters. This suggests that these phenotypes may share some common underlying causes.

19. The rescue is only partial in Figure 8. Could the authors try to perturb the binding sites for *non-stop* in *ex* and *merlin* to more directly test if it binding to these sites to promote border cell migration? Or confirm in their own hands the ChIP seq results? The 1.8MDa SAGA complex acts as a mediator between DNA-bound transcription activators and the RNA polymerase II machinery. Consequently, *Non-stop* does not bind directly to sequence-specific binding sites that could be targeted in the promoters of *ex* and *mer* to address this question. Our data suggest that *not* functions independently of SAGA and in the fullness of time we hope to identify the SAGA-independent factors that facilitate *Non-stop*'s chromatin binding and function.

Unfortunately, this is not currently practicable due to limited resources and staff availability due to Covid-19. The published ChIP-Seq data are of very high quality, and we believe that repeating the ChIP-Seq would constitute a study in its own right.

Minor comments

20. " Later, F-actin accumulates around the cortex of the cluster, as cells alternate their position in the cluster as they move collectively (Bianco et al., 2007)." No evidence is shown of differential cortical actin accumulation in the first vs second phase of migration proposed by the authors. If this is the case, then the analysis in Fig 3 should be done according to the "phase" of migration.

For clarity, we have amended our description of the stages of migration and distinctive collective migration behaviours (as Point 5, Referee 3).

21. It would help the uninitiated reader if the authors provide some diagrams illustrating the protein complexes and pathways analyzed in the manuscript.

An excellent idea. We have included a schematic illustration as suggested, see Fig.9.

Reviewer #3 (Comments to the Authors (Required)):

The manuscript by Badmos and colleagues describes the identification of the Usp22 homolog Non-stop (Not) as a new regulator of border cell collective migration in the *Drosophila* ovary. Not is a deubiquitinating enzyme (DUB) that functions in the histone H2B DUB module of the SAGA transcriptional coactivator complex. The authors first show that not is required for border cell migration, with most clusters failing to reach the oocyte by the correct stage. In some cases, the cluster splits apart. In addition, there is an increase in the number of polar cells with a corresponding increase in border cell number. While several markers of border cell fate were unchanged in not mutant border cells, F-actin localization was altered. Normally, F-actin is primarily localized to the cluster periphery (cortex), but in not mutant clusters more F-actin is now enriched at cell-cell junctions between border cells inside the cluster. Live imaging of not mutant clusters showed that border cells had less polarized protrusions and possibly early cluster "tumbling", suggesting less polarized motility. Not was recently shown to regulate the Arp2/3 and WAVE regulatory complex member Scar. However, Scar mutants have distinct phenotypes from not mutants, suggesting that these two proteins function independently in border cells. Not mutants have similar phenotypes to mutants in the Hippo pathway (Lucas et al., J Cell Biol 2013), including effects on F-actin and polar cell number. The authors propose that Not regulates the upstream Hippo pathway components Expanded and Merlin, and that this is independent of the SAGA HAT module member Ada2b. Interestingly, other polarity proteins such as Crumbs are mislocalized in not mutant border cells. Not may regulate expression of expanded and merlin, as Not protein binds to the transcription start site of both genes and impacts their expression in border cells. Finally, the authors overexpressed Expanded and a known downstream target of the Hippo pathway in border cells, Capping protein B (CPB), and demonstrated partial rescue of border cell migration and polarization of the cluster.

Overall, this manuscript presents new findings on the role of Usp22/Not in regulating

polarized collective cell migration, with new connections to the Hippo pathway through regulation of Expanded and Merlin. There are implications for the polarized migration of cell collectives. This manuscript is generally well-written and well-documented. Most of the data is quantified very clearly. However, some conclusions need additional support, with clarification of key findings.

1. In Figure 1I and J, can the authors clarify if split clusters occur typically when more cells are mosaic mutant for not within the cluster, or when fewer cells are mutant? In other words, are not mutant cells splitting from wild-type cells or from other not mutant cells.?

We have looked at this question in some detail and made the following observations from reanalysing previous image data as well as conducting additional experiments:

- i) Frequency of split clusters is not dependent on size of cluster – some large clusters do not split whereas smaller ones do, and *vice versa* (Fig.2F).
- ii) Splitting was not observed when there was only a very low proportion of mutant cells/cluster (see Fig.2F) – suggesting the cluster integrity is not compromised by the presence of very low numbers of mutant cells.
- iii) We never observed all mutant cells to be in a split cluster; mutant cells in the lagging cluster remain attached to each other or wild type cells;
- iv) Splitting can occur when all the cells in the cluster are mutant, but this is not a necessary outcome;

Based on these observations, we conclude that mutant cells do not have an intrinsic difference in affinity to each other and wild type cells. Quantification of Armadillo staining shows that Arm distribution is not affected by *non-stop* (see Referee 3's point 7). Although, we cannot rule out there being direct effects on other junctional proteins, we consider it likely that the altered F-actin distribution we have observed leads to weakened junctions that are unable to maintain adhesion when put under stochastic mechanical stresses as the cluster moves through the restricted space of the germline material. In this regard, it has previously been noted that collective-level active actomyosin contraction contributes to maintaining the adherence of migrating cells to each other (Chen et al. eLife 2020).

2. The authors state (e.g. p. 2) that *non-stop* is "at the top of a regulatory network underlying collective migration." I am unsure that their data really show this.

We have amended the text to de-emphasise this statement.

3. In Figure 3, the data are convincing and the histograms in panel B are useful. Can the authors show a line in panel A images indicating where made line scans shown in panel B? Panel C should provide statistics similar to what is shown in Figure 8L.

We have made the changes recommended by the reviewer.

4. Videos S1-S3, can the authors add time stamps and/or indicate the length of the video (and what is the frame speed)?

Time stamps have been added to the videos, and frame speed is given in the legend.

5. In the section starting on p. 4 (including Figure 4 results), the authors analyze their movies for protrusions and tumbling. The authors define the two phases of migration as the "initial polarized phase, and a second phase that utilizes collective migration.... cells alternate their position in the cluster as they move collectively." I

disagree that the first phase is not collective, as the cluster is polarized as an entire entity with a single leading protrusion. I would suggest that the authors can just simply define the two phases, both of which have distinct collective behaviors.

We agree that our definition is unclear and have amended the text accordingly.

6. Similarly, it would be helpful if the authors could further clarify what they consider to be "tumbling" behavior (and add this to the Methods). Border cells are known to switch places within the cluster at different times during migration (Prasad and Montell, *Dev Cell* 2007), but rotation or tumbling I believe is more at the cluster level with the entire cell group rotating around an axis (Bianco et al., *Nature* 2007; Poukkula et al., *J Cell Biol* 2011). The videos they show do not really illustrate this very clearly, especially the wild-type video S2. I am wondering if the behavior they are seeing in not mutant clusters is cell-cell exchange rather than premature tumbling? Moreover, do they see differences in these cluster behaviors if the not mutant cells partially migrated versus didn't migrate at all (such as shown in video S3)?

As stated in the Methods (see also Law *et al.*, *JCB* 2013), we defined early tumbling as the mean percentage of frames per time lapse movie that showed rounded clusters, exhibiting changes in the position of individual cells within the cluster for two or more consecutive frames in the first half of migration. We agree with the reviewer that this behaviour is not always clear in clusters that have failed to detach from the epithelium, as in our original example. To illustrate this more clearly, we have included an additional example (Video S5) where the mutant cells have partially migrated.

7. For Figure 6 data, I am convinced by changes in the patterns and/or levels of ex-lacZ, Crumbs, and aPKC in not mutant clusters versus wild type. I agree that Ena looks mostly similar between not mutants and wild type. However, the changes in Merlin are not very obvious in the images shown and should be quantified. Likewise, any changes to the Armadillo localization are not that clear. All of the data in this figure should be quantified as much as possible, and N's reported as the authors do in other figures within this manuscript. It appears that the Crumbs mean intensity is reported in Figure 8K - something similar for the rest of these markers could be included to be more convincing.

We have performed line scans of border cells stained with markers shown in Fig.6 and plotted the mean intensity at the front middle and back of the cluster as we have done elsewhere in the manuscript. The referee is right to say that the changes to Armadillo localisation are quite subtle – in fact, when we carefully quantified Arm we did not see any statistical difference in the distribution of staining intensity (Fig.6P). We have amended the text to reflect this. Since we did not have many examples of Merlin staining as we only possessed a limited stock of gifted antibody, we decided to repeat these experiments with an endogenously-expressed, fluorescently tagged form of Merlin that we obtained from Rick Fehon (University of Chicago). The YFP-Mer that we used is fully functional, as judged by the ability to rescue null *mer* mutations (Su *et al* *Dev Cell* 2018). We found that, similar to anti-Merlin staining, YFP-Merlin levels were reduced in border cells homozygous for *non-stop* mutant. Importantly, detection of the YFP-Merlin enabled a more robust measurement of the effect of *not1* loss of function in mutant clones marked with lacZ (see Fig. 6F-H).

8. The authors find that Not binds to the expanded and merlin promoters (ChIPSeq from Li et al., 2017; Figure 7A and 7B). Did the authors check for Not ChIPSeq to the

start sites of other gene members of the Hippo pathway, including Crumbs? We have revisited this issue with reference to the ChIP-Seq data from Li et al. As well as *mer* and *ex*, there are Not binding sites in many of the hippo pathway gene promoters, including: *hpo*, *kib*, *zyx*, *crb* (Table S1). It is therefore intriguing to speculate that multiple components of the hippo pathway may be functionally affected by *non-stop* loss of function, although it should be noted (as Referee 2 comments), that inputs and outputs of the hippo pathway are complex and redundant. Whilst beyond the scope of our current study, it will be interesting to look at the regulation of other hippo pathway genes more carefully in different contexts in the fullness of time. We have commented on this issue in the revised text as a point that may be of wider interest to the field. It should be noted however, that *Ada2b* shares the ability to bind hippo pathway genes, except *ex* and *mer* (Table S1), so we consider *ex* and *mer* to be the relevant targets of *not* in the context of border cell migration.

9. The authors show that *Ada2b*, a SAGA-specific HAT module subunit is not required for border cell migration nor impacts *ex-lacZ* or F-actin. How definitive is this data, having only used one mutant allele? Is it possible to test another component of the HAT module to confirm this result? Also, in Figure 7C, is *ex-lacZ* higher in mutant cells?

We have quantified the data in Figure 7 and included data from additional experiments described below to strengthen the conclusion that there is a HAT-independent requirement for *non-stop* in border cell migration. Our quantification of *ex-lacZ* indicates that the level is modestly elevated in *ada2b¹* clones (Fig7D), but this has no impact on the distribution of F-actin or Crb (Fig7F). We were able to obtain RNAi lines to explore the role of several other HAT components (*gcn5*, *sgf29*, *ada3*), including a well-characterised line for *gcn5* that strongly abrogates its histone acetyltransferase activity (Carre et al. Mol Cell Biol 2005). RNAi-knockdown of *gcn5*, had no effect on actin polarity (Fig.7F, H) or the distribution of Crb (Fig.7F, I). Furthermore, knockdown of neither *gcn5*, *sgf29* or *ada3* (at 25°C or 30°C) had a significant effect on border cell migration (Fig7G).

10. In the last section of the results (p. 8), the authors conclude that "expanded is a critical transcriptional target of non-stop required for its function in border cells." I would argue that their data support expanded (and Cpb) as being targets/downstream, but either there are other parallel downstream targets or technical reasons that neither Expanded nor Cpb strongly rescue the not mutant phenotypes. There is only mild rescue of migration (Figure 8M; the authors should add statistics here). From their data, it seems that Expanded has more impact on rescuing Crumbs localization (Figure 8G and K), whereas Cpb seems to have a greater rescue on F-actin localization (Figure 8J and L).

The reviewer is right to point out that the rescue of *not¹* is incomplete. For clarity, we have amended the discussion to clarify that other parallel downstream targets may exist. Statistics have been added to Figure 8. We agree with the reviewer that *expanded* has more impact on Crumbs, whereas *cpb* has more impact on F-actin localisation. This might suggest that re-expression of *Mer*, or indeed other genes, with *Ex* is required to restore a more normal F-actin distribution. For technical reasons this is not something we could readily attempt at this time.

11. Figure 8L, the authors could show statistics comparing the phenotypes of the rescues to the not mutant border cells, not just to wild type.

These statistics are now included in the revised Figure 8.

12. It is intriguing how not mutants strongly impact Crumbs localization and F-actin in border cell clusters. In the discussion (p. 10 "non-stop regulates the distribution of polarity determinants"), they mention possible relationship to Moesin. This may be beyond the scope of this manuscript, but did the authors look at Moesin localization in not mutants (e.g. Moesin localization as shown in Ramel et al. Nat Cell Biol 2013)? The authors discuss a recent paper that showed that Moesin stabilizes Crumbs (Aguilar-Aragon et al., 2020). Likewise, Ramel et al. (2013) showed that Moesin regulates F-actin organization and polarity of the border cell cluster. Could an effect on Moesin explain both the Crumbs and F-actin effects by Non-stop?

We have now done these experiments using an antibody that was kindly provided to us by Greg Emery (University of Montreal). Neither Moe distribution or levels were affected in *not¹* mutant border cells (Fig.S2).

Minor comment:

1. Missing call out to figure on p. 5 "Polarisation of the polarity determinant Crb...." (Figure 5E).

We have inserted a reference to the relevant figure.

On-Target Details

ID	Source	Seq len	Target gene	OTE	Target	Len	CDS	Align	Region	Match
45776	VDRC GD	333	not, FBgn0013717	11	not-RC	2989	639 2129	213 545	5'UTR	333bp perfect match
			all isoform(s)		not-RB	2488	639 2129	213 545	5'UTR	333bp perfect match

Off-Target Details

Gene	Transcript	Match Length	Mismatches	Gaps	Identity	e-value	Bits	Predicted function
CG16791	CG16791-RD	23	0	0	100.0	4.0E-4	46.1	Unknown
CG16791	CG16791-RC	23	0	0	100.0	4.0E-4	46.1	
CG16791	CG16791-RB	23	0	0	100.0	4.0E-4	46.1	
CG16791	CG16791-RA	23	0	0	100.0	4.0E-4	46.1	
mRNA-cap	mRNA-cap-RA	21	0	0	100.0	0.0060	42.1	RNGTT orthologue; mRNA guanylyltransferase activity
stx	stx-RA	21	0	0	100.0	0.0060	42.1	Midnolin homologue expressed in brain cell body
stx	stx-RB	21	0	0	100.0	0.0060	42.1	
AstA	AstA-RA	21	0	0	100.0	0.0060	42.1	Involved in neuropeptide signaling pathway expressed in the nervous system
CG43102	CG43102-RC	19	0	0	100.0	0.094	38.2	Predicted Rho guanyl-nucleotide exchange factor activity expressed in primary trachea and spermatozoon
CG43102	CG43102-RB	19	0	0	100.0	0.094	38.2	
CG43102	CG43102-RA	19	0	0	100.0	0.094	38.2	
CG43102	CG43102-RF	19	0	0	100.0	0.094	38.2	
CG43102	CG43102-RG	19	0	0	100.0	0.094	38.2	
CG43689	CG43689-RF	22	0	0	100.0	0.0020	44.1	

CG43689	CG43689-RG	22	0	0	100.0	0.0020	44.1	Predicted DNA-binding transcription repressor activity; alleles display phenotypes in myofibril; Z disc; sarcomere; mesothoracic tergum
CG43689	CG43689-RI	22	0	0	100.0	0.0020	44.1	
CG43689	CG43689-RH	22	0	0	100.0	0.0020	44.1	
CG43689	CG43689-RE	22	0	0	100.0	0.0020	44.1	
btsz	btsz-RG	21	0	0	100.0	0.0060	42.1	Synaptotagmin family protein, predicted to have Rab GTPase binding activity
btsz	btsz-RJ	21	0	0	100.0	0.0060	42.1	
CG31475	CG31475-RB	21	0	0	100.0	0.0060	42.1	Predicted to be involved in calcium-ion regulated exocytosis
CG31475	CG31475-RB	18	0	0	100.0	0.37	36.2	
CG31475	CG31475-RA	21	0	0	100.0	0.0060	42.1	
CG31475	CG31475-RA	18	0	0	100.0	0.37	36.2	
bun	bun-RA	19	0	0	100.0	0.094	38.2	Probable transcription factor required for peripheral nervous system morphogenesis, eye development and oogenesis.
bun	bun-RP	19	0	0	100.0	0.094	38.2	
bun	bun-RG	19	0	0	100.0	0.094	38.2	
bun	bun-RF	19	0	0	100.0	0.094	38.2	
CG13643	CG13643-RB	21	0	0	100.0	0.0060	42.1	Predicted to be involved in chitin metabolic process
CG13643	CG13643-RD	21	0	0	100.0	0.0060	42.1	
CG13643	CG13643-RE	21	0	0	100.0	0.0060	42.1	
CG13643	CG13643-RC	21	0	0	100.0	0.0060	42.1	

March 24, 2021

Re: JCB manuscript #202007005R

Dr. Daimark Bennett
University of Liverpool
Molecular Physiology and Cell Signalling
Institute of Systems Molecular and Integrative Biology
Crown Street
Liverpool L69 7ZB
United Kingdom

Dear Dr. Bennett,

Thank you for submitting your revised manuscript entitled "Drosophila USP22/non-stop polarises the actin cytoskeleton during collective border cell migration." The manuscript has been seen by the original reviewers whose full comments are appended below. While the reviewers continue to be overall positive about the work in terms of its suitability for JCB, some important issues remain.

You will see that Reviewers #1&2 point out that descriptions of phenotypes of not1 rescues by Ex and Scar in the text do not match the statistical analyses and ask to revise text so that it accurately reflects the data and modify conclusions accordingly. Reviewer #2 also notes that not1 mutants produce extra polar and border cells whereas Not CRISPR knockout does not and asks either to validate by RNAi or to tone down conclusions regarding autonomy of the extra polar cell phenotype. We encourage you to address this question with new data if possible.

Our general policy is that papers are considered through only one revision cycle; however, given that the suggested changes are relatively minor we are open to one additional short round of revision.

Please submit the final revision within one month, along with a cover letter that includes a point by point response to the remaining reviewer comments.

Thank you for this interesting contribution to Journal of Cell Biology. You can contact me or the scientific editor listed below at the journal office with any questions, cellbio@rockefeller.edu or call (212) 327-8588.

Sincerely,

Anna Huttenlocher, MD
Monitoring Editor
Journal of Cell Biology

Dan Simon, PhD
Scientific Editor
Journal of Cell Biology

Reviewer #1 (Comments to the Authors (Required)):

I am satisfied with the revised manuscript, which is now suitable for publication in JCB.

Reviewer #2 (Comments to the Authors (Required)):

The authors have added quantifications of the phenotypes and in general made an effort to address the reviewers' questions and concerns. While the revised manuscript is somewhat improved, the statistical analyses provided do not always match what is written in the text, there is still uncertainty regarding the autonomy of the extra polar cell phenotype, and the model that the authors provide is not fully supported by the data provided. The following issues should be addressed prior to publication:

What is clear in this manuscript is that Not is required autonomously within migratory border cells for their migration as shown in Figure 1. To improve the presentation, please indicate on the figure what all the colors mean in panels A-I (and throughout the manuscript) so the reader does not have to go to the figure legend to know what is shown. What is GFP in these panels?

The text states: "Clusters containing only not1 outer BCs showed strong migration defects (Fig.1G,H) and this was not significantly enhanced by the presence of not1 polar cells (compare >50% or >80% not1 outer BC, to >50% and 100% not1 cluster, respectively, Fig1H)." But they did not compare >50% not1 outer BC, to >50% not1 cluster in Fig 1H, which would be the appropriate comparison to support the conclusion.

The not1 mutant clones produce extra polar and border cells (Fig 2B), but the CRISPR approach does not (Fig S1B). Can the authors provide an explanation? One concern is that this CRISPR approach has not been validated in this biological context (post-mitotic follicle cells are polyploid and this might impact the effectiveness of the technique). If the CRISPR technique is effective, then a major concern is that the conclusion regarding not1 in polar cells might not be correct. The authors have all the stocks necessary to use not RNAi to knockdown Not in polar cells using upd-Gal4. Although there would be concern about the possible off-target effect, the RNAi-resistant rescue construct used elsewhere in the paper could address this concern. This would be an important addition to the evidence that polar cell knockdown does not cause extra polar cells. It is also possible that the extra polar cell phenotype is a consequence of a defect early in development at the precursor stage. If the authors cannot obtain conclusive evidence, perhaps they could tone down the conclusion regarding autonomy and mention the outstanding possibilities.

The most important issues for the authors to address are the data analysis, interpretation, and conclusions regarding rescue of the not1 phenotype by Ex and Scar.

The text states: "However, there was a modest enrichment of Scar in the junctions between outer BCs of not1 clusters, accompanying the increased F-actin at this location (Fig.5E)". Please change "modest" to "significant though modest," because there is a statistically significant difference. If the authors wish to indicate that the differences between them are not biologically meaningful, then they should indicate why they think statistically significant results are not meaningful, whereas results that do not rise to statistical significance are considered important (see below regarding Ex overexpression). Perhaps the authors should re-consider their interpretations.

Also in the results section regarding the effect of ada2b on Ex-lacZ and migration, please again change "modest" to "significant though modest".

Can the authors comment on whether *ada2b*, like *gcn5* is required for polarity of Crbs?
A main conclusion is that: "Overexpression of *ex* partially rescues cell migration and polarity defects". And they state: "Strikingly, *ex* overexpression (*ex+*) substantially restored more normal Crb and F-actin distributions in *not1* BCs (Fig.8G-H and K-L) and significantly suppressed the effect of *not1* on migration (Fig.8M; the mean percentage migration of *ex+* *not1* BC clusters was 57.4 {plus minus}3.2%, compared to 38.7 {plus minus}2.9%, for *not1* alone).

However, the migration defects are not significantly different upon overexpression of *Ex* in *not1* mutants compared to *not1* mutants alone. This is a concern because it is a major conclusion and in the proposed model.

The figure shows that neither *Ex* nor *cpb* significantly rescues migration though they do rescue Crb and actin phenotypes. Please rewrite the text to accurately reflect the data.

The model includes proteins that were not tested in this manuscript and therefore over-states what can actually be concluded.

Additional comments

Did the authors mean to cite figure 2F for the 1.7 fold increase in border cells (first paragraph from results section, "*not1* regulates polar cell number...". The figure 2I is cited and this seems incorrect. More generally the graphs shown in Figure 2 panels E and F are quite unusual and confusing and don't really support the conclusion drawn from them.

In the text under the section "*not1* regulates polar cell number non-cell autonomously" - Figure 1E should read 2E.

Figure S2 needs to be annotated.

In Fig 3 comparisons should not be made between migrating clusters and those that have completed migration. Please include a wild type stage 10 with complete migration, and then compare the experimental results to either the mid-migration or docked control depending on the location of the cluster in the experimental sample. Or at least show there is no difference in signal between control migratory and control docked clusters.

In figure 4 it would be good to either bin by detached vs attached or to only quantify clusters that have delaminated as protrusion dynamics can differ greatly during detachment vs early migration. Differences could represent additional time attached in the experimental as opposed to differences in dynamics during migration, and this analysis could rule out that possibility.

"Premature tumbling". Is there evidence of the tumbling being premature - as in a normal biological process activated earlier than normal, as opposed to tumbling just being a phenotype unrelated to the normal process?

Reviewer #3 (Comments to the Authors (Required)):

In this revised manuscript, Badmos and co-authors address the function of non-stop, a new regulator of collective cell migration. Specifically, they demonstrate a SAGA-independent role for non-stop in polarizing the border cell cluster, likely through the Hippo pathway components Expanded and Merlin. This study will be of broad interest, especially those who study collective cell migration and cell polarity. The authors have substantially revised their manuscript and have addressed all of my concerns. In particular, they provided further evidence that other HAT module subunits are not required, provided additional quantification of their data, along with other clarifications of their data and conclusions. I appreciate the efforts the authors made to address all of the reviewers' concerns; I am satisfied with these revisions.

Two minor suggestions:

1) in the last part of the results, the authors state "Stikingly, ex overexpression (ex+) substantially restored more normal Crb and F-actin... and significantly suppressed the effect of not[1] on migration (Fig. 8M)..." The statistics show that the rescue of migration is not "significant" though from the percent migration differences, their data do support "partial suppression." I would suggest a text re-wording to be more careful.

2) Figure S2A - to be visually clearer, it would help to have a box drawn around the inset of border cells.

2nd Revision - Authors' Response to Reviewers: April 7, 2021

We would like to thank the reviewers again for their time and excellent suggestions for improvement of the manuscript. Our point-by-point response can be found copied below. In addition to the recommended changes, we have made a few typographical corrections to the manuscript (e.g. mcherry to mCherry; *Scar* to *scar*). All changes can be found marked in the Related Manuscript File.

March 24, 2021

Re: JCB manuscript #202007005R

Dr. Daimark Bennett
University of Liverpool
Molecular Physiology and Cell Signalling
Institute of Systems Molecular and Integrative Biology
Crown Street
Liverpool L69 7ZB
United Kingdom

Dear Dr. Bennett,

Thank you for submitting your revised manuscript entitled "Drosophila USP22/non-stop polarises the actin cytoskeleton during collective border cell migration." The manuscript has been seen by the original reviewers whose full comments are appended below. While the reviewers continue to be overall positive about the work in terms of its suitability for JCB, some important issues remain.

You will see that Reviewers #1&2 point out that descriptions of phenotypes of not1 rescues by Ex and Scar in the text do not match the statistical analyses and ask to revise text so that it accurately reflects the data and modify conclusions accordingly. Reviewer #2 also notes that not1 mutants produce extra polar and border cells whereas Not CRISPR knockout does not and asks either to validate by RNAi or to tone down conclusions regarding autonomy of the extra polar cell phenotype. We encourage you to address this question with new data if possible.

Our general policy is that papers are considered through only one revision cycle; however, given that the suggested changes are relatively minor we are open to one additional short round of revision.

Please submit the final revision within one month, along with a cover letter that includes a point by point response to the remaining reviewer comments.

Thank you for this interesting contribution to Journal of Cell Biology. You can contact me or the scientific editor listed below at the journal office with any questions, cellbio@rockefeller.edu or call (212) 327-8588.

Sincerely,

Anna Huttenlocher, MD
Monitoring Editor
Journal of Cell Biology

Dan Simon, PhD
Scientific Editor
Journal of Cell Biology

Reviewer #1 (Comments to the Authors (Required)):

I am satisfied with the revised manuscript, which is now suitable for publication in JCB.
There are no issues to address.

Reviewer #2 (Comments to the Authors (Required)):

The authors have added quantifications of the phenotypes and in general made an effort to address the reviewers' questions and concerns. While the revised manuscript is somewhat improved, the statistical analyses provided do not always match what is written in the text, there is still uncertainty regarding the autonomy of the extra polar cell phenotype, and the model that the authors provide is not fully supported by the data provided. The following issues should be addressed prior to publication:

What is clear in this manuscript is that Not is required autonomously within migratory border cells for their migration as shown in Figure 1. To improve the presentation, please indicate on the figure what all the colors mean in panels A-I (and throughout the manuscript) so the reader does not have to go to the figure legend to know what is shown. What is GFP in these panels?

We have added labels to Figure 1, and other figures where appropriate, to make it easier for readers to understand at a glance what is shown. In some of the later figures we have not labelled GFP and DNA on certain panels because we felt additional labels would clutter the figure and because the colour scheme for these channels is ubiquitous throughout the manuscript. In panels Fig.1 A-I, GFP is in green.

The text states: "Clusters containing only not1 outer BCs showed strong migration defects (Fig.1G,H) and this was not significantly enhanced by the presence of not1 polar cells (compare >50% or >80% not1 outer BC, to >50% and 100% not1 cluster, respectively, Fig1H)." But they did not compare >50% not1 outer BC, to >50% not1 cluster in Fig 1H, which would be the appropriate comparison to support the conclusion.

We have added this comparison, which shows there is no significant difference between >50% *not¹* outer BC, to >50% *not¹* cluster, supporting our conclusion.

The not1 mutant clones produce extra polar and border cells (Fig 2B), but the CRISPR approach does not (Fig S1B). Can the authors provide an explanation? One concern is that this CRISPR approach has not been validated in this biological context (post-mitotic follicle cells are polyploid and this might impact the effectiveness of the technique). If the CRISPR technique is effective, then a major concern is that the conclusion regarding not1 in polar

cells might not be correct. The authors have all the stocks necessary to use not RNAi to knockdown Not in polar cells using upd-Gal4. Although there would be concern about the possible off-target effect, the RNAi-resistant rescue construct used elsewhere in the paper could address this concern. This would be an important addition to the evidence that polar cell knockdown does not cause extra polar cells. It is also possible that the extra polar cell phenotype is a consequence of a defect early in development at the precursor stage. If the authors cannot obtain conclusive evidence, perhaps they could tone down the conclusion regarding autonomy and mention the outstanding possibilities.

A key difference between the clonal and CRISPR experiments is that *not¹* clones were evident at a much earlier point in development and were not restricted to the border cell cluster. Therefore, the most likely explanation for the difference here is that the extra polar cell phenotype is a consequence of a defect early in development in a non-identical population of cells. Even though the RNAi does not offer a clean loss of function, we agree that the *upd-GAL4* experiment that the reviewer suggested may have helped to further confirm the non-cell autonomous requirement for *not* in preventing extra polar cells. However, unfortunately when we attempted this we found that *upd>not^{IR}* animals were lethal at 25°C. We have also trialed growing these flies at lower temperature to dampen GAL4 expression but this failed to yield survivors. Therefore, we have done as the reviewer has suggested and toned down the conclusion regarding autonomy and clarified our interpretation with regard to *not*'s requirement. In the Results the text now reads: "However, extra polar cells occurred in clusters containing wild type polar cells (Fig.2C,D), indicating that the effect on polar cell number may be non-cell autonomous. Targeting *not* in polar or outer border cells by CRISPR-Cas9 did not significantly affect cell number (Fig.S1B). This may be because of differences in the efficacy of inducing loss-of-function, or because there is a requirement for *not* at an earlier point in egg chamber development that is disrupted in *not¹* clones."

The most important issues for the authors to address are the data analysis, interpretation, and conclusions regarding rescue of the *not1* phenotype by Ex and Scar.

The text states: "However, there was a modest enrichment of Scar in the junctions between outer BCs of *not1* clusters, accompanying the increased F-actin at this location (Fig.5E)". Please change "modest" to "significant though modest," because there is a statistically significant difference. If the authors wish to indicate that the differences between them are not biologically meaningful, then they should indicate why they think statistically significant results are not meaningful, whereas results that do not rise to statistical significance are considered important (see below regarding Ex overexpression). Perhaps the authors should re-consider their interpretations.

We agree with the reviewers' suggestions with regard to the description of *scar* experiments, and have changed "modest" to "significant though modest,". However, this does not affect the model we proposed, in which we have indicated a role for Scar in supporting the formation of F-actin in junctions upon *not* loss of function (Results: "In summary, rather than being depleted in *not¹* clusters Scar is slightly enriched in BC-BC junctions where it contributes to F-actin formation"; see also Fig.9 where Scar is depicted).

Also in the results section regarding the effect of *ada2b* on Ex-lacZ and migration, please again change "modest" to "significant though modest".

We have made these changes to the revised manuscript.

Can the authors comment on whether *ada2b*, like *gcn5* is required for polarity of Crbs?

gcn5 is not required for Crb polarity (Fig. 7F). We did not test the effect of *ada2b* on Crbs.

A main conclusion is that: "Overexpression of *ex* partially rescues cell migration and polarity defects". And they state: "Strikingly, *ex* overexpression (*ex*+) substantially restored more normal Crb and F-actin distributions in *not1* BCs (Fig.8G-H and K-L) and significantly suppressed the effect of *not1* on migration (Fig.8M; the mean percentage migration of *ex*+ *not1* BC clusters was 57.4 {plus minus}3.2%, compared to 38.7 {plus minus}2.9%, for *not1* alone). However, the migration defects are not significantly different upon overexpression of *Ex* in *not1* mutants compared to *not1* mutants alone. This is a concern because it is a major conclusion and in the proposed model.

The figure shows that neither *Ex* nor *cpb* significantly rescues migration though they do rescue Crb and actin phenotypes.

Please rewrite the text to accurately reflect the data.

Contrary to what the reviewer has said here, we found that *cpb*⁺ did significantly rescue migration of *not1*^l clusters, (though not to wild type levels, see Fig.8M).

The reviewer is right to point out that *ex*⁺ does not significantly rescue migration. We are grateful that the reviewer spotted this – we had omitted to update our description of the *ex*⁺ results in the light of our revised data. This has now been corrected in the revision.

This issue is also raised by reviewer 3 who said: "The statistics show that the rescue of migration is not "significant" though from the percent migration differences, their data do support "partial suppression."” We agree.

Our interpretation of this is that other genes, including perhaps its binding partner Merlin, are also required. The reason why *cpb*⁺ is able to significantly rescue migration is likely because it is able to restore near normal F-actin polarity (Fig.8L, no significant difference to wild type). These points are discussed in the Conclusion.

The model includes proteins that were not tested in this manuscript and therefore over-states what can actually be concluded.

We have removed labels to proteins that were not directly tested to avoid the suggestion that the model over-states what can be concluded.

Additional comments

Did the authors mean to cite figure 2F for the 1.7 fold increase in border cells (first paragraph from results section, "*not1* regulates polar cell number...". The figure 2I is cited and this seems incorrect. More generally the graphs shown in Figure 2 panels E and F are quite unusual and confusing and don't really support the conclusion drawn from them.

In the text under the section "*not1* regulates polar cell number non-cell autonomously" - Figure 1E should read 2E.

We thank the reviewer for spotting these typographical errors. Figure 2I was cited in error. We have deleted this call out – the correct citation is provided at the end of the sentence (Fig.2B). We have also corrected 1E to 2E. Neither Reviewer 1 or 3 raised an issue regarding clarity of Figures 2 panels E and F. We believe these panels provide an effective way to show the relationship between % mutant cells, size of cluster (total nuclei) and % migration while unambiguously presenting all the datapoints. However, the description of these panels was somewhat abbreviated due to reasons of space and we have therefore amended the text as follows to highlight the key observations.

"The presence of extra BCs might impact on function. However, strong migration defects are clearly observed in *not1*^l mutant clusters with normal numbers of cells (6-8 cells) (Fig.2E); at best there was a weak negative correlation between total number of nuclei and % migration

($R=-0.43$). Cell splitting was also not limited to larger clones containing extra cells (Fig.2F) and there was not a strong correlation between cluster size and frequency of splitting ($R=0.01$).”

Figure S2 needs to be annotated.

We have included a box around panel S2A and annotated the figure.

In Fig 3 comparisons should not be made between migrating clusters and those that have completed migration. Please include a wild type stage 10 with complete migration, and then compare the experimental results to either the mid-migration or docked control depending on the location of the cluster in the experimental sample. Or at least show there is no difference in signal between control migratory and control docked clusters.

We have included an image of a wild type stage 10 egg chamber showing complete migration, along with quantification of F-actin staining using line scans, which shows no significant difference in signal between control migratory and control docked clusters. A representative line scan is shown in Fig.3 panel B and quantitation of repeats is shown in panel C, along with statistical comparisons.

In figure 4 it would be good to either bin by detached vs attached or to only quantify clusters that have delaminated as protrusion dynamics can differ greatly during detachment vs early migration. Differences could represent additional time attached in the experimental as opposed to differences in dynamics during migration, and this analysis could rule out that possibility.

Delamination of *not¹* clusters was impaired in all of the time-lapse images quantified in Fig.4 and therefore it would not be appropriate to bin these examples by detached vs attached. We have clarified this in the legend to Figure 4. However, to address this possibility we have quantified the example shown in Supplementary Video S5, where the cluster does readily delaminate. In this example we find that the protrusion dynamics closely resemble those of attached *not¹* clusters reported in Fig.4. The ratio of Front, Side, Back protrusions is 9.5, 62.0, 28.5 whereas the *not¹* mean distribution in “attached” samples (Fig.4) is 15.7, 62.4, 21.9; the tumbling index is 100% (n=62 frames). Therefore, at least in this example of a detached early migrating cluster (Video S5) we do not see different protrusion dynamics. To aid comparisons, we have provided the distribution of protrusions for the Supplementary videos (S3-5) in the legends. To acknowledge the possibility that some differences may be accounted for by the attachment status, we have added the following caveat in the results, also pointing readers to the supplementary video discussed above:

“Since *not¹* clusters often failed to delaminate we cannot rule out that some differences in protrusion dynamics result from a failure to detach from the epithelium, although we did observe similar protrusion behaviour in a detached cluster that we imaged (Video S5).”

“Premature tumbling”. Is there evidence of the tumbling being premature - as in a normal biological process activated earlier than normal, as opposed to tumbling just being a phenotype unrelated to the normal process?

The reviewer is correct – we can’t distinguish between these possibilities and have therefore removed the word “premature” from the text.

Reviewer #3 (Comments to the Authors (Required)):

In this revised manuscript, Badmos and co-authors address the function of non-stop, a new regulator of collective cell migration. Specifically, they demonstrate a SAGA-independent role for non-stop in polarizing the border cell cluster, likely through the Hippo pathway components Expanded and Merlin. This study will be of broad interest, especially those who study collective cell migration and cell polarity. The authors have substantially revised their manuscript and have addressed all of my concerns. In particular, they provided further evidence that other HAT module subunits are not required, provided additional quantification of their data, along with other clarifications of their data and conclusions. I appreciate the efforts the authors made to address all of the reviewers' concerns; I am satisfied with these revisions.

Two minor suggestions:

1) in the last part of the results, the authors state "Stikingly, ex overexpression (ex+) substantially restored more normal Crb and F-actin.... and significantly suppressed the effect of not[1] on migration (Fig. 8M)..." The statistics show that the rescue of migration is not "significant" though from the percent migration differences, their data do support "partial suppression." I would suggest a text re-wording to be more careful.

We are grateful that the reviewer spotted this – we had omitted to update the text in the light of our revised data. This has now corrected in the revision.

2) Figure S2A - to be visually clearer, it would help to have a box drawn around the inset of border cells.

We have included a box around panel S2A and annotated the figure for clarity.

April 14, 2021

RE: JCB Manuscript #202007005RR

Dr. Daimark Bennett
University of Liverpool
Molecular Physiology and Cell Signalling
Institute of Systems Molecular and Integrative Biology
Crown Street
Liverpool L69 7ZB
United Kingdom

Dear Dr. Bennett,

Thank you for submitting your revised manuscript entitled "Drosophila USP22/non-stop polarises the actin cytoskeleton during collective border cell migration". We would be happy to publish your paper in JCB pending final revisions necessary to address one final minor Reviewer comment and to meet our formatting guidelines (see details below).

A. MANUSCRIPT ORGANIZATION AND FORMATTING:

Full guidelines are available on our Instructions for Authors page, <https://jcb.rupress.org/submission-guidelines#revised>. **Submission of a paper that does not conform to JCB guidelines will delay the acceptance of your manuscript.**

- 1) Text limits: Character count for Articles is < 40,000, not including spaces. Count includes title page, abstract, introduction, results, discussion, and acknowledgments. Count does not include materials and methods, figure legends, references, tables, or supplemental legends.
- 2) Figures limits: Articles may have up to 10 main text figures.
- 3) Figure formatting: Scale bars must be present on all microscopy images, including inset magnifications. Molecular weight or nucleic acid size markers must be included on all gel electrophoresis. You currently do not have scale bars in Figures 4A,B and 7H,I.
- 4) Statistical analysis: Error bars on graphic representations of numerical data must be clearly described in the figure legend. The number of independent data points (n) represented in a graph must be indicated in the legend. Statistical methods should be explained in full in the materials and methods. For figures presenting pooled data the statistical measure should be defined in the figure legends. Please also be sure to indicate the statistical tests used in each of your experiments (both in the figure legend itself and in a separate methods section) as well as the parameters of the test (for example, if you ran a t-test, please indicate if it was one- or two-sided, etc.). Also, if you used parametric tests, please indicate if the data distribution was tested for normality (and if so, how). If not, you must state something to the effect that "Data distribution was assumed to be normal but

this was not formally tested."

5) Abstract: We suggest splitting up the following sentence into two instead of using a semicolon - "Here we identify, in a genetic screen for deubiquitinating enzymes involved in border cell migration, an essential role for non-stop/USP22 in the expression of Hippo pathway components expanded and merlin; loss of non-stop function consequently leads to a redistribution of F-actin and the polarity determinant Crumbs, loss of polarised actin protrusions and tumbling of the border cell cluster."

6) Materials and methods: Should be comprehensive and not simply reference a previous publication for details on how an experiment was performed. Please provide full descriptions (at least in brief) in the text for readers who may not have access to referenced manuscripts. The text should not refer to methods "...as previously described."

7) Please be sure to provide the sequences for all of your primers/oligos and RNAi constructs in the materials and methods. You must also indicate in the methods the source, species, and catalog numbers (where appropriate) for all of your antibodies.

8) Microscope image acquisition: The following information must be provided about the acquisition and processing of images:

- a. Make and model of microscope
- b. Type, magnification, and numerical aperture of the objective lenses
- c. Temperature
- d. Imaging medium
- e. Fluorochromes
- f. Camera make and model
- g. Acquisition software
- h. Any software used for image processing subsequent to data acquisition. Please include details and types of operations involved (e.g., type of deconvolution, 3D reconstitutions, surface or volume rendering, gamma adjustments, etc.).

10) Supplemental materials: There are strict limits on the allowable amount of supplemental data. Articles may have up to 5 supplemental figures and 10 videos. Please also note that tables, like figures, should be provided as individual, editable files. A summary of all supplemental material should appear at the end of the Materials and methods section.

11) eTOC summary: A ~40-50 word summary that describes the context and significance of the findings for a general readership should be included on the title page. The statement should be written in the present tense and refer to the work in the third person. It should begin with "First author name(s) et al..." to match our preferred style.

13) A separate author contribution section is required following the Acknowledgments in all research manuscripts. All authors should be mentioned and designated by their first and middle initials and full surnames. We encourage use of the CRediT nomenclature (<https://casrai.org/credit/>).

14) ORCID IDs: ORCID IDs are unique identifiers allowing researchers to create a record of their various scholarly contributions in a single place. At resubmission of your final files, please consider providing an ORCID ID for as many contributing authors as possible.

B. FINAL FILES:

-- High-resolution figure and video files: See our detailed guidelines for preparing your production-ready images, <https://jcb.rupress.org/fig-vid-guidelines>.

-- Cover images: If you have any striking images related to this story, we would be happy to consider them for inclusion on the journal cover. Submitted images may also be chosen for highlighting on the journal table of contents or JCB homepage carousel. Images should be uploaded as TIFF or EPS files and must be at least 300 dpi resolution. Please also provide a brief description.

Thank you for this interesting contribution, we look forward to publishing your paper in Journal of Cell Biology.

Sincerely,

Anna Huttenlocher, MD
Monitoring Editor
Journal of Cell Biology

Dan Simon, PhD
Scientific Editor
Journal of Cell Biology

Reviewer #2 (Comments to the Authors (Required)):

The authors have responded constructively to the remaining criticisms and the manuscript is now suitable for publication. The only small thing to clarify (in the figure legend) is what the source of GFP is in Figure 1. Is it UAS-GFP?